# Combined Rho-kinase inhibition and immunogenic cell death triggers and propagates immunity against cancer

Gi-Hoon Nam[1,2], Eun Jung Lee[2,3], Yoon Kyoung Kim[1,2], Yeonsun Hong[1,2], Yoonjeong Choi[1,2], Myung-Jeom Ryu[2,4], Jiwan Woo[5], Yakdol Cho[5], Dong June Ahn[1], Yoosoo Yang[2], Ick-Chan Kwon[1,2], Seung-Yoon Park[2,6] & In-San Kim[1,2]

Activation of T cell immune response is critical for the therapeutic efficacy of cancer immunotherapy. Current immunotherapies have shown remarkable clinical success against several cancers; however, significant responses remain restricted to a minority of patients. Here, we show a therapeutic strategy that combines enhancing the phagocytic activity of antigen-presenting cells with immunogenic cell death to trigger efficient antitumour immunity. Rho-kinase (ROCK) blockade increases cancer cell phagocytosis and induces antitumour immunity through enhancement of T cell priming by dendritic cells (DCs), leading to suppression of tumour growth in syngeneic tumour models. Combining ROCK blockade with immunogenic chemotherapy leads to increased DC maturation and synergistic CD8[+] cytotoxic T cell priming and infiltration into tumours. This therapeutic strategy effectively suppresses tumour growth and improves overall survival in a genetic mouse mammary tumour virus/Neu tumour model. Collectively, these results suggest that boosting intrinsic cancer immunity using immunogenic killing and enhanced phagocytosis is a promising therapeutic strategy for cancer immunotherapy.

[1] KU-KIST Graduate School of Converging Science and Technology, Korea University, Seoul 02841, Republic of Korea. [2] Center for Theragnosis, Biomedical Research Institute, Korea Institute Science and Technology (KIST), Seoul 02792, Republic of Korea. [3] Department of Chemical Engineering, School of Applied Chemical Engineering, Kyungpook National University, Daegu 41566, Republic of Korea. [4] College of Medicine, Yonsei University, Seoul 120-752, Republic of Korea. [5] Research Animal Resource Center, Korea Institute of Science and Technology (KIST), Seoul 02792, Republic of Korea. [6] Department of Biochemistry, School of Medicine, Dongguk University, Gyeongju 38066, Republic of Korea. These authors contributed equally: Gi-Hoon Nam, Eun Jung Lee. Correspondence and requests for materials should be addressed to S.-Y.P. (email: psyoon@dongguk.ac.kr) or to I.-S.K. (email: iskim14@kist.re.kr)

Reinforcement of intrinsic immune responses is an important factor that contributes to the therapeutic efficacy of cancer immunotherapy, an anticancer approach that is currently undergoing a revolution[1]. Eliciting effective tumour antigen-specific immunity requires targeting the initial stages of the anticancer immunity cycle, including tumour antigen release, uptake and presentation and T cell priming. Several molecular targets have been singled out in efforts to modulate tumour cell phagocytosis. For example, anti-CD20 monoclonal antibody has been found to simulate phagocytosis of malignant B cells[2] and drive antitumour immune responses[3]. However, therapeutic strategies targeting cancer cells may have limited applications because their therapeutic efficacy is dependent on the expression of specific target molecules in cancer cells. Therefore, it may be necessary to potentiate the function of antigen-presenting cells (APCs) at the initial stages of the anticancer immunity cycle using strategies that target host immune cells.

The small GTPase RhoA and its downstream signalling effectors play important roles in the organization and dynamics of the actin cytoskeleton in many biological processes, including cell adhesion and migration[4,5]. Rho-associated kinases (ROCKs), which are key downstream effectors of RhoA, have been implicated in tumour motility, invasion and growth[6]. Several studies have demonstrated therapeutic benefits of ROCK blockade on tumour cell migration and metastasis in a variety of tumour models[7–10]. RhoA/ROCK signalling has also been implicated in extracellular matrix (ECM) remodelling and tissue stiffness, which are associated with tumour aggressiveness[11,12]. A recent study has shown that antitumour effect of ROCK blockade is linked to FasL overexpression and T cell-mediated immune response[13]. In addition, RhoA/ROCK signalling was found to negatively regulate the engulfment of apoptotic cells[14,15]. Accordingly, blockade of the RhoA/ROCK pathway using a ROCK inhibitor increases the phagocytic capacity of macrophages and enhances their clearance of apoptotic cells[14,16]. These observations suggest the possibility that ROCK blockade promotes tumour cell phagocytosis by APCs, thereby leading to processing of cancer-specific antigens and activation of T cell immunity against cancer.

Tumour cells are antigenic, reflecting the abundance of somatic mutations in their genome; however, their immunogenicity in terms of eliciting cytotoxic T cell responses is relatively low because processes involved in host immunity activation, such as antigen presentation, take place in an immunosuppressive tumour environment[17]. Depending on the initiating stimulus, cancer cell death can be immunogenic or non-immunogenic[18]. Some chemotherapeutics, such as doxorubicin (Dox), mitoxantrone and oxaliplatin, have been reported to induce immunogenic cell death (ICD) of cancer cells, leading to activation of antitumour immune responses[19–21]. However, a previous study showed that the effect of ICD inducers is independent of adaptive immunity in some spontaneous mammary tumour models[22], suggesting that ICD inducers may not be sufficient to induce effective antitumour immunity. These reports prompted us to hypothesize that immunogenic killing of tumour cells using an ICD inducer in conjunction with a phagocytosis enhancer might be a suitable combined antitumour immunotherapy for effectively 'awakening' intrinsic tumour immunity. Here, we show that ROCK blockade reduces tumour growth through increased cancer cell phagocytosis as well as T cell priming. Furthermore, the combination of an ICD inducer and ROCK blockade markedly induces effective antitumour immunity and suppresses tumour progression in syngeneic tumour models as well as a genetically engineered model.

## Results

### ROCK blockade enhances cancer cell clearance by phagocytes.

As a first step in testing our combined treatment strategy, we investigated whether blockade of ROCK enhances engulfment of cancer cells by phagocytes. Macrophages and DCs were differentiated from bone marrow cells, as assessed by flow cytometry for F4/80 (macrophages) and CD11c (DCs) expression on the cell surface (Supplementary Fig. 1). Treatment of bone marrow-derived macrophages (BMDMs) or bone marrow-derived dendritic cells (BMDCs) with the ROCK inhibitor, Y27632, led to a significant increase in the engulfment of CT26.CL25 colon cancer cells and B16F10-Ova melanoma cells (Fig. 1a, c). The percentages of phagocytosis were substantially increased in the Y27632-treated BMDMs (9.37 and 8.43) and BMDCs (9.99 and 10.09) as compared to the vehicle-treated BMDMs (3.76 and 3.1) and BMDCs (2.92 and 4.52), respectively (Fig. 1b, d). To definitely confirm engulfment of cancer cells, we labelled CT26.CL25 cells with the pH-sensitive dye, pHrodo-succinimidyl ester (pHrodo-SE), which emits red fluorescence in acidic phagosomes[23], and co-cultured with BMDMs or BMDCs. In agreement with the results from flow cytometry, ROCK blockade led to a substantial increase in red fluorescent cells engulfed by BMDMs and BMDCs (Fig. 1e–g). The phagocytosis-promoting effect of Y27632 was reversible, as assessed by phagocytosis assays after Y27632 washout (Supplementary Fig. 3). A previous study showed that the blockade of RhoA/ROCK signalling induces phosphatidylserine (PS) exposure on the cell surface[24], which acts as an 'eat me' signal for phagocytic clearance. Accordingly, we addressed the possibility that the surface exposure of PS induced by Y27632 increases the phagocytosis of cancer cells. A significant change in PS exposure by Y27632 was not observed in both cells even if cancer cells were treated with Y27632 for 24 h (Supplementary Fig. 2). To further determine whether the phagocytosis-promoting effect of Y27632 is attributable to modulation of the ability of BMDMs and BMDCs to phagocytose cancer cells, we treated phagocytes or cancer cells with Y27632 and performed phagocytosis assays. We found that treatment of phagocytes with Y27632 promoted cancer cell phagocytosis, whereas treatment of cancer cells with Y27632 had no effect on their engulfment (Fig. 1h and Supplementary Fig. 4), suggesting that ROCK blockade augments phagocytosis by modulating phagocytes rather than by acting on tumour cells.

RhoA/ROCK pathway has been found to modulate the regulatory myosin light chain (MLC) by direct phosphorylation or by inhibiting MLC phosphatase, leading to actomyosin assembly and cell contraction[25,26]. It is possible that decrease of contractility by ROCK blockade facilitates cell shape change for effective phagocytosis. We analysed the effect of inhibition of the motor protein myosin II on cancer cell phagocytosis. Treatment of BMDMs or BMDCs with the myosin II inhibitor, blebbistatin, led to a significant increase in the engulfment of CT26.CL25 and B16F10-Ova cells (Fig. 1i, j and Supplementary Fig. 5), reminiscent of that observed for phagocytes treated with Y27632. These results suggest that cancer cell phagocytosis is modulated by RhoA/ROCK/myosin II pathway.

### Phagocytes are crucial to antitumour effect of ROCK blockade.

We next assessed the effects of Y27632 on tumour growth in two subcutaneous syngeneic tumour models: CT26.CL25 colon cancer and B16F10-Ova melanoma. Tumour cells were transplanted subcutaneously into the flank of 8-week-old male BALB/c or C57BL/6 mice. After tumours had reached a size of ~50–100 mm³, mice were intravenously (i.v.) administered Y27632 (Supplementary Fig. 6a, b). Y27632 treatment substantially decreased tumour volume compared with vehicle treatment (Fig. 2a, b). When tumours were isolated and weighed at the end of the

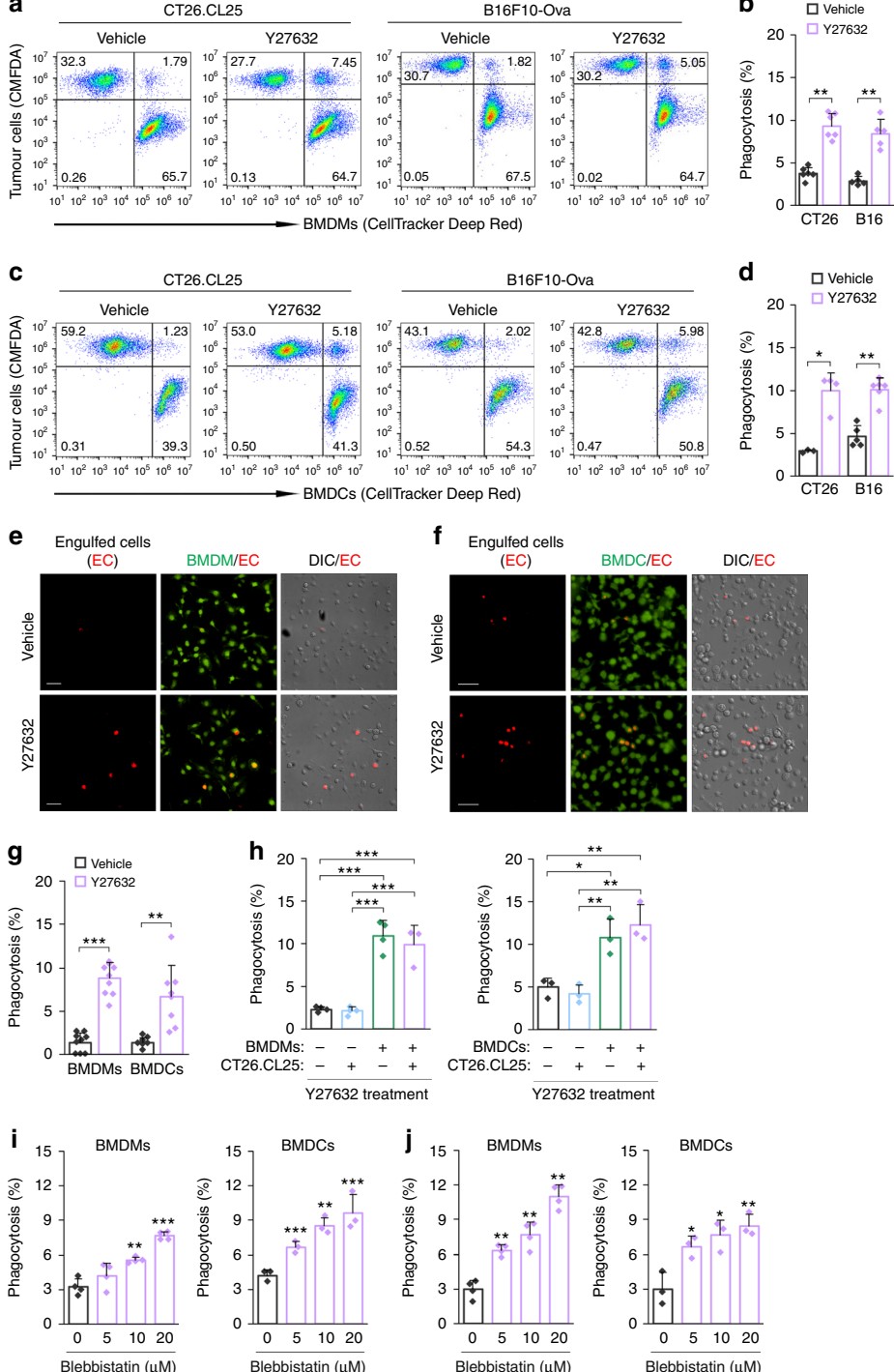

**Fig. 1** ROCK blockade enhances cancer cell clearance by phagocytes. **a–d** CellTracker Deep Red-labelled BMDMs (**a**, **b**) or BMDCs (**c**, **d**) were co-cultured with CMFDA-stained CT26.CL25 or B16F10-Ova cells for 2 h in the presence of Y27632 (30 μM) or vehicle (PBS), and then analysed by flow cytometry. Upper right quadrants indicate BMDMs or BMDCs harbouring cancer cells. **a**, **c** Representative flow cytometry plots. **b**, **d** Phagocytosis, calculated as the percentage of the total number of BMDMs or BMDCs containing cancer cells. Data are presented as means ± s.d. ($n$ = 3–6). **e–g** CMFDA-labelled BMDMs (**e**) or BMDCs (**f**) were co-cultured with pHrodo-labelled CT26.CL25 cells for 2 h in the presence of Y27632 or vehicle, and analysed under a fluorescent microscope. **e**, **f** Representative images of BMDMs (**e**) and BMDCs (**f**) phagocytosing pHrodo-labelled CT26.CL25 cells. CMFDA-labelled BMDMs or BMDCs are shown in green, and engulfed CT26.CL25 cells are shown in red. Scale bar, 50 μm. **g** Phagocytosis, calculated as the percentage of the total number of BMDMs or BMDCs containing cancer cells. Data are presented as means ± s.d. ($n$ = 8–10). **h** CellTracker-labelled BMDMs (left panel) or BMDCs (right panel) were co-cultured with CMFDA-stained CT26.CL25 cells for 2 h under the indicated conditions, and phagocytosis (%) was analysed by flow cytometry. Data are presented as means ± s.d. ($n$ = 3–4). **i**, **j** CellTracker-labelled BMDMs (left panels) or BMDCs (right panels) were co-cultured with CMFDA-stained CT26.CL25 (**i**) and B16F10-Ova cells (**j**) for 2 h in the presence of blebbistatin (5–20 μM) or vehicle, and phagocytosis (%) was analysed by flow cytometry. Data are presented as means ± s.d. ($n$ = 3–4). $^*P < 0.05$, $^{**}P < 0.01$, $^{***}P < 0.001$; significance was determined by Mann–Whitney test (**b**, **d**, **g**), one-way ANOVA with Tukey's post hoc test (**h**) or unpaired Student's $t$ test (**i**, **j**)

experiment, Y27632-treated mice had smaller tumour burthen than vehicle-treated mice (Fig. 2c). No substantial difference in body weight was observed between Y27632-treated and vehicle-treated groups (Supplementary Fig. 7). To determine if the antitumour activity associated with ROCK blockade was mediated by phagocytes, we generated a CT26.CL25 cell line expressing mCherry and treated CT26.CL25-mCherry tumour-bearing mice with clodronate liposomes (Supplementary Fig. 6c), which selectively induce the death of phagocytes[27]. Injection of clodronate liposomes reduced the percentages of F4/80+ macrophages and CD11c+ DCs to 3.1% and 26%, respectively (Supplementary Fig. 8), and robustly abrogated the antitumour response produced by systemic administration of Y27632 (Fig. 2d). To explore the effect of Y27632 on phagocytosis of cancer cells in vivo, we analysed the uptake of CL26.CL25-mCherry cells by macrophages and DCs in tumour tissues from CT26.CL25-mCherry tumour-bearing mice. Notably, the percentages of mCherry+ macrophages and DCs were approximately twofold higher in Y27632-treated mice than in vehicle-treated mice (Fig. 2e, f), demonstrating the phagocytosis-promoting effect of ROCK blockade in vivo. In addition, we found that Y27632 increased the phagocytic activity in macrophages and DCs from mice that are not bearing tumour (Supplementary Fig. 9). To assess whether Y27632 promotes an immune response against tumour-associated antigens, we isolated tumour-draining lymph node (LN) cells from CT26.CL25-mCherry tumour-bearing mice, and analysed their immune response against β-galactosidase (β-gal), a model tumour-associated antigen in CT26.CL25 cells[28]. Tumour-draining LN cells from Y27632-treated mice produced more interferon-γ (IFN-γ) in response to a β-gal peptide than those from vehicle-treated mice, an effect that was abrogated by phagocyte depletion (Fig. 2g). To assess whether cytotoxic CD8+ T cells are responsible for IFN-γ production in the tumour-draining LN cells from Y27632-treated mice, we isolated CD8+ and CD8− cells from the tumour-draining LN cells and analysed IFN-γ production against β-gal peptide. Notably, Y27632-mediated IFN-γ production was found in CD8+ cells (Fig. 2h). These results suggest that increased uptake of tumour antigen by ROCK blockade potentiates antitumour T cell immunity.

**ROCK blockade induces antitumour immunity**. To assess the role of T cell-mediated immunity in the antitumour activity of Y27632, we transplanted CT26.CL25 cells into syngeneic BALB/c nude mice and analysed the therapeutic effect of systemic Y27632 treatment (Supplementary Fig. 6d). Y27632 treatment had a modest tumour-suppressing effect in nude mice (Fig. 3a), suggesting the importance of T cell immunity in Y27632-mediated antitumour effects. We then analysed the effects of Y27632 on tumour growth in the CT26.CL25 tumour model in which CD4+ or CD8+ T cells were depleted by intraperitoneal injection of a neutralizing antibody (Supplementary Figs. 6e and 10). Depletion of CD8+ T cells impaired the antitumour activity of Y27632, as evidenced by the more rapid progression of tumours, whereas CD4+ T cell depletion had no effect (Fig. 3b, c). These results indicate that CD8+ T cells are required for Y27632-mediated tumour regression. To determine whether Y27632 provided protective immune memory, we re-challenged mice in which the primary tumour was surgically removed after Y27632 treatment by injecting a higher number of tumour cells in the contralateral flank (Supplementary Fig. 6f). The percentage of tumour-free mice was substantially higher in the Y27632-treated group than in the vehicle-treated group (Fig. 3d), suggesting that ROCK blockade contributes to the induction of durable systemic immune memory and prevention of tumour relapse.

**ROCK blockade increases DC-mediated T cell priming**. We further assessed the mechanisms through which Y27632 promotes T cell responses against tumour-associated antigens. Tumour cell phagocytosis by APCs can lead to migration of antigen-loaded APCs to tumour-draining LNs, where they fully mature and prime antitumour CD8+ T cell responses[29,30]. To assess DC maturation following Y27632 treatment in vivo, we analysed expression of costimulatory ligands in DCs from tumour-draining LNs of B16F10-Ova tumour-bearing mice. CD40 and CD86 expression were comparable in DCs from tumour-bearing mice regardless of Y27632 treatment (Fig. 4a and Supplementary Fig. 11a–c). Similar results were observed when BMDCs pretreated with Y27632 were co-cultured with CT26.CL25 cells (Supplementary Fig. 11d–f). Next, to assess whether ROCK blockade increases cross-priming of cytotoxic CD8+ T cells, we performed adoptive transfer (i.v.) of carboxy-fluorescein succinimidyl ester (CFSE)-labelled ovalbumin (OVA)-specific T (OT-I) cells into B16F10-Ova tumour-bearing mice and measured proliferation of OT-I T cells. Notably, the percentage of proliferating OT-I T cells was markedly increased in tumour-draining LNs from Y27632-treated mice compared with that in vehicle-treated mice (Fig. 4b, c). To determine the ability of DCs and macrophages to prime antitumour T cell responses following Y27632 treatment ex vivo, we magnetically enriched DCs and macrophages from tumours or tumour-draining LNs from B16F10-Ova tumour-bearing mice and then co-cultured them with OVA-specific OT-I cells. DCs from tumours or tumour-draining LNs from Y27632-treated mice increased T cell activation, as assessed by IFN-γ production, whereas macrophages did not (Fig. 4d, e); co-culture with DCs from Y27632-treated mice also increased the number of IFN-γ-producing CD8+ T cells (Fig. 4f). These results suggest that T cell priming by DCs is important for eliciting an antitumour immune response after Y27632 therapy.

To further assess the mechanism by which ROCK blockade increases T cell priming, we isolated CD11c+ DCs from tumour-draining LN and tumours in B16F10-Ova tumour-bearing mice and analysed DC subtypes and their ability for cross-presentation of tumour-specific antigen. The percentage of CD103+ DC was substantially increased in both tumour-draining LNs and tumours from Y27632-treated mice, compared with those from vehicle-treated mice (Fig. 4g and Supplementary Fig. 12). Cross-presentation of OVA-derived peptide to major histocompatibility complex-1 (MHC-I) was increased in CD103+ and CD8+ DCs from tumour-draining LNs and CD103+ DCs from tumours of Y27632-treated mice (Fig. 4h and Supplementary Fig. 13). Because migratory CD103+ DC is the main intratumoural myeloid population that transports tumour-specific antigens to the tumour-draining LNs[31,32], we isolated CD11c+ DCs in tumours from B16F10-Ova tumour-bearing mice and analysed phagocytic ability of CD103+ and CD11b+ DCs for B16F10-Ova cells. Tumour cell phagocytosis was found to be increased in CD103+ DCs, but not CD11b+ DCs, from Y27632-treated mice (Fig. 4i and Supplementary Fig. 14). Furthermore, although we did not detect a significant increase in the expression of costimulatory factors in total CD11c+ DCs, CD40 expression was markedly increased in CD103+ DC population from the tumour-draining LNs from Y27632-treated mice (Fig. 4j and Supplementary Fig. 15). These results suggest that enhanced phagocytosis of cancer cell-derived antigens by migratory CD103+ DCs could increase their maturation and cross-presentation of tumour antigen, leading to T cell priming.

**ROCK blockade enhances processing of Dox-treated cells**. Anthracyclines, such as Dox, have been shown to induce ICD in

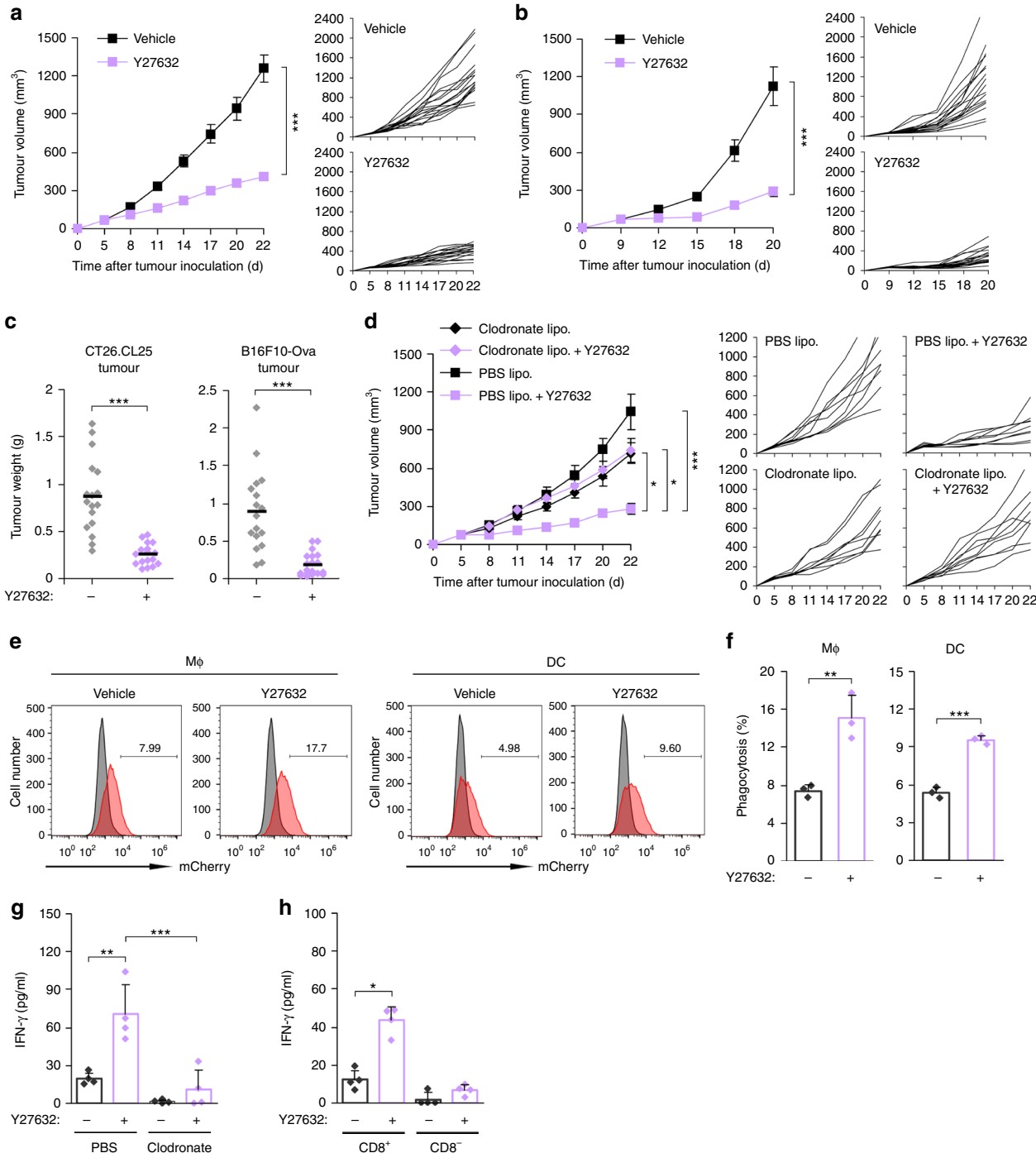

**Fig. 2** Phagocytes are crucial to antitumour effect of ROCK blockade. **a–c** CT26.CL25 tumour-bearing BALB/c mice (**a**) and B16F10-Ova tumour-bearing C57BL/6 mice (**b**) were injected (i.v.) with Y27632 (10 mg kg$^{-1}$) or vehicle (PBS). Tumour dimensions were measured at the indicated times using calipers and tumour volume was calculated (mm$^3$). Data are presented as means ± s.e.m. (left panels) and tumour growth curves of individual mice (right panels) (CT26.CL25 model, $n = 17$; B16F10-Ova model, $n = 17$–19). **c** Tumour weights in individual mice were analysed at the end of experiments. Each dot represents tumour weight of individual mice. Bars indicate mean values. **d–g** CT26.CL25-mCherry tumour-bearing BALB/c mice were injected (i.v.) with Y27632 (10 mg kg$^{-1}$) together with clodronate liposomes or PBS liposomes as described in Supplementary Fig. 6c. **d** Tumour volume was measured at the indicated time. Data are presented as means ± s.e.m. (left panel) and tumour growth curves of individual mice (right panels) ($n = 9$). **e, f** For the PBS liposomes-treated group, F4/80$^+$ macrophages or CD11c$^+$ DCs were isolated from tumours, and the percentage of cells harbouring mCherry$^+$ tumour cells was determined by flow cytometry. **e** Representative flow cytometry plots. The grey peaks represent mCherry levels in macrophages and DCs from CT26.CL25 tumour-bearing mice (mCherry-negative tumour). **f** Data are presented as means ± s.d. ($n = 3$). **g** At 22 days after tumour inoculation, single-cell suspensions from tumour-draining LNs were stimulated with β-gal-derived peptide for 48 h, and secreted IFN-γ was measured by ELISA. Data are presented as means ± s.d. ($n = 4$). **h** At 22 days after tumour inoculation, CD8$^+$ and CD8$^-$ cells were isolated from tumour-draining LNs and stimulated with β-gal-derived peptide for 48 h, and secreted IFN-γ was measured by ELISA. Data are presented as means ± s.d. ($n = 4$). *$P < 0.05$, **$P < 0.01$, ***$P < 0.001$; significance determined by unpaired Student's $t$ test (**a, f**), Mann–Whitney test (**b, c**) or one-way ANOVA with Tukey's post hoc test (**d, g, h**)

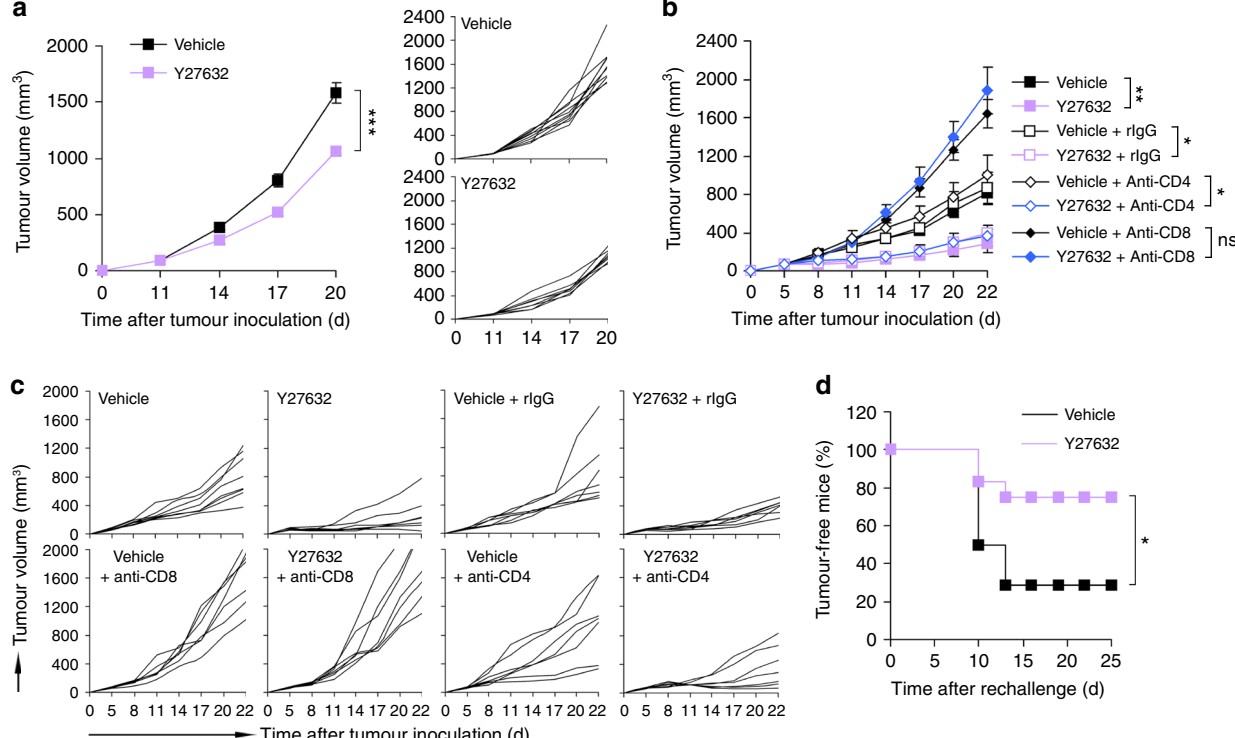

**Fig. 3** ROCK blockade induces antitumour immunity. **a** CT26.CL25 tumour-bearing nude mice were injected (i.v.) with Y27632 (10 mg kg$^{-1}$) or vehicle. Tumour volume was measured at the indicated times. Data are presented as means ± s.e.m. (left panel) and tumour growth curves of individual mice (right panels) (*n* = 8–10). **b, c** BALB/c mice were transplanted with CT26.CL25 cells and injected with Y27632 (10 mg kg$^{-1}$) at the indicated times, as described in Supplementary Fig. 6e. Five days after tumour inoculation, mice were injected (i.p.) with neutralizing anti-CD4 or anti-CD8 antibody. **b** Tumour volume was measured in each group at the indicated times. Data are presented as means ± s.e.m. and **c** tumour growth curves of individual mice (*n* = 7–8). **d** CT26.CL25 tumour-bearing BALB/c mice were injected (i.v.) with Y27632 or vehicle. Twenty-two days after tumour inoculation, primary tumours were surgically removed from mice. Seven days later, a high dose of tumour cells (7 × 10$^6$ cells) was introduced into the contralateral flank, and the percentage of tumour-free mice was evaluated at the indicated times (*n* = 12–14). *$P$ < 0.05, **$P$ < 0.01, ***$P$ < 0.001, ns, not significant; significance determined by unpaired Student's *t* test (**a, b**) or log-rank test (**d**)

cancer cells, thereby eliciting an effective antitumour immune response[19,21]. Accordingly, we assessed whether ROCK blockade augmented DC-mediated phagocytosis of cells dying an immunogenic death. Treatment of BMDCs with Y27632 caused a substantial increase in the engulfment of Dox-treated tumour cells, as assessed by flow cytometry (Fig. 5a, b). Similar results were observed when BMDCs pretreated with Y27632 were co-cultured with pHrodo-labelled, Dox-treated CT26.CL25 cells in the presence of Y27632 (Fig. 5c, d). ROCK blockade also increased DC-mediated phagocytosis of cells dying a non-immunogenic or necrotic cell death (Supplementary Fig. 16). Cells undergoing ICD have been shown to increase DC maturation and cross-presentation of tumour antigen[33,34]. We therefore assessed the effect of ROCK blockade on these processes using our co-culture system. Y27632 treatment increased CD40 and CD86 expression in DCs co-cultured with Dox-treated cells by 19% and 37%, respectively, compared with vehicle treatment (Fig. 5e, f and Supplementary Fig. 17). Cross-presentation of an OVA-derived peptide to MHC-I was markedly increased by co-culture of BMDCs with Dox-treated B16F10-Ova cells in the presence of Y27632, as assessed by staining for the SIINFEKL-H-2k$^b$ complex on the DC surface (Fig. 5g and Supplementary Fig. 18). These results suggest that the combination of an ICD inducer and ROCK blockade efficiently increases the phagocytosis of tumour cells and processing of engulfed tumour antigens.

**Combined therapy effectively induces antitumour immunity.**
To assess the therapeutic efficacy of the combination of Dox and

Y27632 on tumour growth, we administered B16F10-Ova tumour-bearing mice with Y27632 alone or in combination with Dox (Supplementary Fig. 19). We found that the combination of Dox and Y27632 exerted a more dramatic tumour-suppressing effect than Dox or Y27632 alone, producing complete tumour regression in 25% (3 of 12) of mice (Fig. 6a, b). To understand the mechanisms underlying the observed therapeutic benefit, we analysed the sequential immunological events elicited by uptake of tumour-associated antigen necessary to achieve adaptive antitumour immunity. Monotherapy with Y27632 or Dox did not affect DC maturation and cross-presentation of OVA-specific antigen in vivo, as assessed by flow cytometry for costimulatory ligands and the SIINFEKL-H-2k$^b$ complex on the DC surface, respectively (Fig. 6c, d). In contrast, the combination of Y27632 and Dox increased CD40 and CD86 expression as well as cross-presentation of OVA-derived peptide to MHC-I in DCs from tumour-draining LNs of B16F10-Ova tumour-bearing mice (Fig. 6c, d and Supplementary Fig. 20). To assess whether combined therapy potentiates the ability of DCs to prime CD8$^+$ T cells ex vivo, we incubated DCs from tumour-draining LNs with OVA-specific OT-I T cells 20 days after tumour inoculation and analysed T cell activation by evaluating IFN-γ production. Notably, combined therapy induced more secretion of IFN-γ than each monotherapy (Fig. 6e), a finding paralleled by the increased proportion of IFN-γ-secreting CD8$^+$ T cells (Fig. 6f), suggesting that combined therapy effectively activated T cell immunity. Next, to assess infiltration of cytotoxic T cells into tumours following Dox and/or Y27632 treatment, we performed

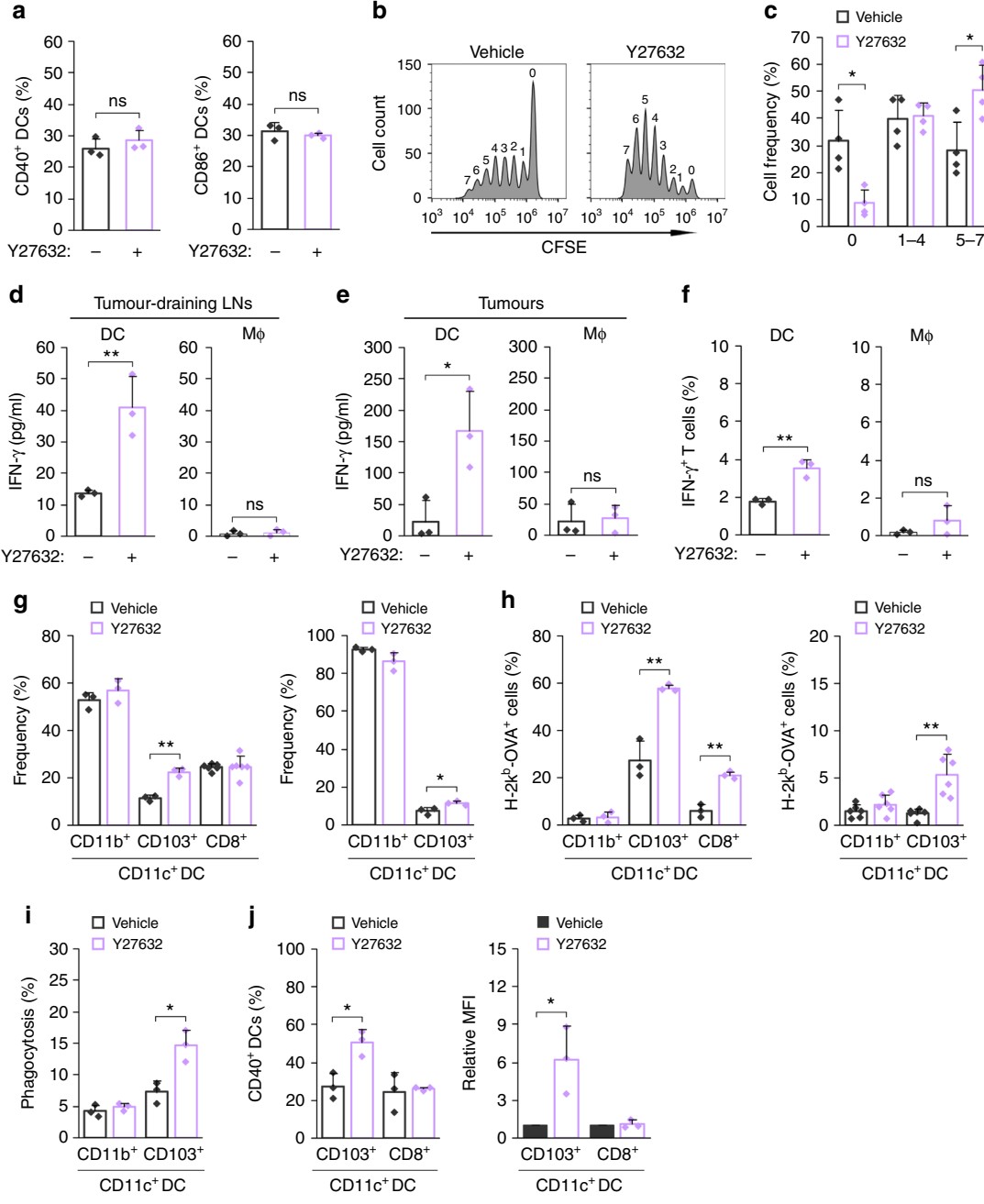

**Fig. 4** ROCK blockade increases DC-mediated T cell priming. **a** The percentage of CD40[+] and CD86[+] DCs in tumour-draining LNs from B16F10-Ova tumour-bearing mice was determined by flow cytometry ($n = 3$). **b, c** B16F10-Ova tumour-bearing mice were injected (i.v.) with Y27632 or vehicle. On day 20, mice were injected (i.v.) with CFSE-labelled OT-I T cells; after 3 days, the proliferation of OVA-specific OT-I T cells in tumour-draining LNs was analysed by flow cytometry. **b** Representative flow cytometry plots. **c** Percentages of the indicated CFSE-labelled OT-I T cells ($n = 4$). **d, e** CD11c[+] DCs or F4/80[+] macrophages from B16F10-Ova tumour-bearing mice treated with Y27632 or vehicle were magnetically isolated from tumour-draining LNs (**d**) or tumours (**e**). DCs or macrophages were incubated with OT-I T cells for 72 h, and secreted IFN-γ was analysed by ELISA ($n = 3$). **f** CD11c[+] DCs or F4/80[+] macrophages were isolated from tumour-draining LNs from B16F10-Ova tumour-bearing mice treated with Y27632 or vehicle and co-cultured with OT-I T cells for 48 h. The percentage of IFN-γ-producing CD8[+] T cells was analysed by flow cytometry ($n = 3$). **g** CD11c[+] DCs from B16F10-Ova tumour-bearing mice treated with Y27632 or vehicle were magnetically isolated from tumour-draining LNs or tumours. The percentages of CD11b[+], CD103[+] and CD8[+] DCs were analysed by flow cytometry ($n = 3$–6). **h** In CD11b[+], CD103[+] and CD8[+] DCs from tumour-draining LNs or tumours, the percentage of H-2k[b]-OVA[+] DCs was analysed by flow cytometry ($n = 3$–6). **i** CD11c[+] DCs were isolated from tumours in B16F10-Ova tumour-bearing BALB/c mice injected (i.v.) with Y27632 (10 mg kg[−1]) and incubated with CMFDA-stained B16F10-Ova for 2 h, and the percentage of CD11b[+] or CD103[+] DCs harbouring tumour cells was determined by flow cytometry ($n = 3$). **j** CD11c[+] DCs were isolated among tumour-draining LNs from B16F10-Ova tumour-bearing BALB/c mice injected (i.v.) with Y27632 (10 mg kg[−1]), and the percentage and relative MFI of CD40[+] cells in CD103[+] or CD8[+] DCs was analysed by flow cytometry ($n = 3$). Data are presented as means ± s.d. (*$P < 0.05$, **$P < 0.01$, ns, not significant; significance determined by unpaired Student's $t$ test)

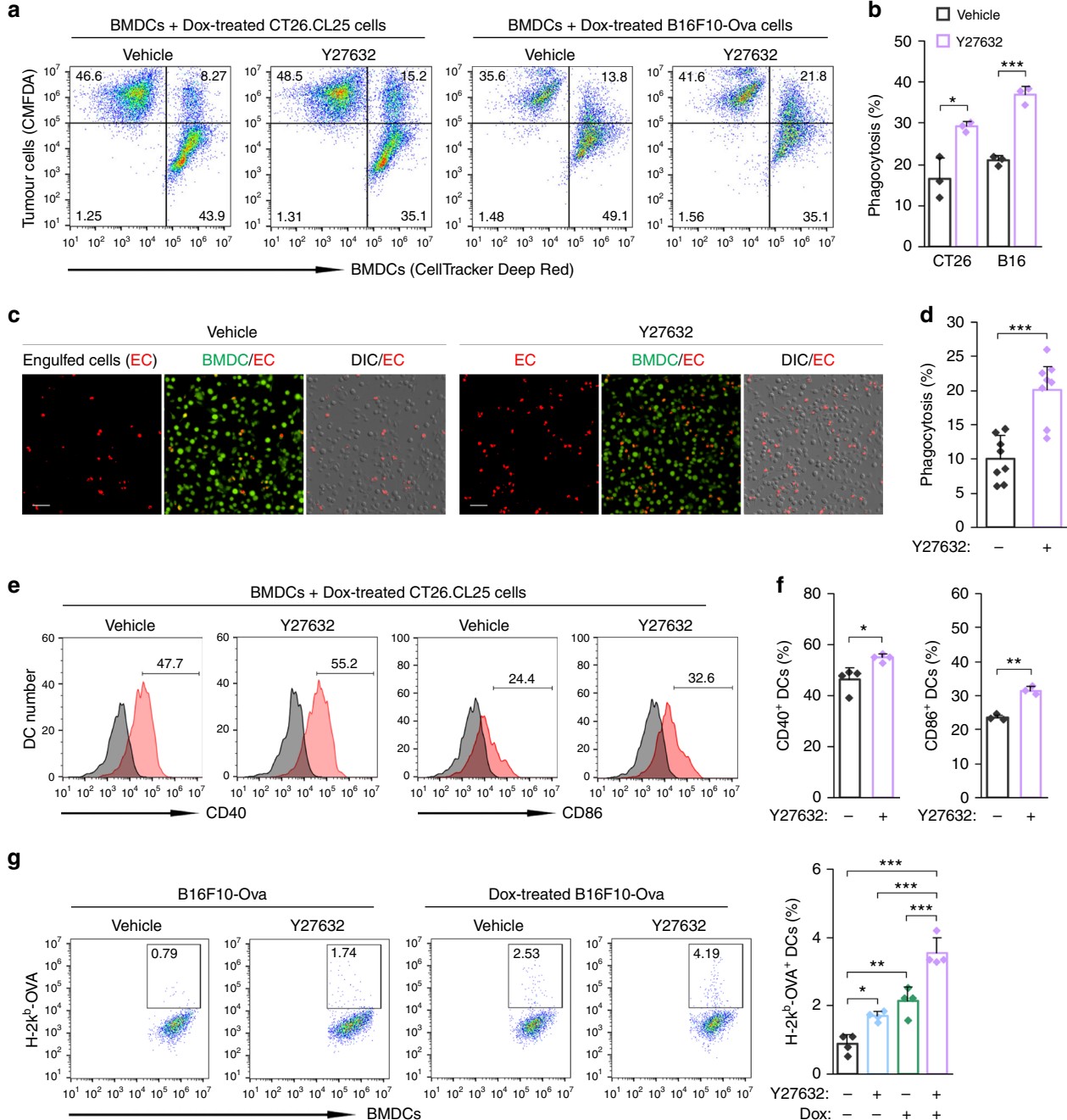

**Fig. 5** ROCK blockade enhances processing of Dox-treated cells. **a**, **b** CellTracker Deep Red-labelled BMDCs were co-cultured with CMFDA-stained, Dox-treated CT26.CL25 or B16F10-Ova cells for 5 min, and then analysed by flow cytometry. **a** Representative flow cytometry plots. Upper right quadrant indicates BMDCs phagocytosing cancer cells. **b** The percentage of phagocytosis. Data are presented as means ± s.d. ($n = 3$). **c**, **d** CMFDA-labelled BMDCs were co-cultured with pHrodo-labelled CT26.CL25 cells for 5 min in the presence of Y27632 or vehicle, and analysed under a fluorescent microscope. **c** Representative images of BMDCs phagocytosing pHrodo-labelled, Dox-treated CT26.CL25 cells. CMFDA-labelled BMDCs are shown in green, and engulfed CT26.CL25 cells are shown in red. Scale bar, 50 μm. **d** Phagocytosis, calculated as the percentage of the total number of BMDCs containing cancer cells. Data are presented as means ± s.d. ($n = 8$). **e**, **f** BMDCs were co-cultured with Dox-treated CT26.CL25 cells for 4 h, and un-engulfed cells were then removed. After further incubation for 20 h, the expressions of CD40 and CD86 in BMDCs were analysed by flow cytometry. **e** Representative flow cytometry plots. The gray peaks represent the isotype control. **f** The percentage of CD40+ and CD86+ BMDCs. Data are presented as means ± s.d. ($n = 3$–4). **g** BMDCs were pretreated with Y27632 or vehicle and co-cultured with B16F10-Ova cells or Dox-treated B16F10-Ova cells for 4 h and then un-engulfed cells were removed. After further incubation for 20 h, the percentage of H-2k$^b$-OVA+ BMDCs was analysed by flow cytometry using an antibody recognizing a complex of the OVA peptide (SIINFEKL) with H-2k$^b$ (MHC-I). Data are presented as means ± s.d. ($n = 4$). $^*P < 0.05$, $^{**}P < 0.01$, $^{***}P < 0.001$; significance determined by unpaired Student's $t$ test (**b**, **d**, **f**), or one-way ANOVA with Tukey's post hoc test (**g**)

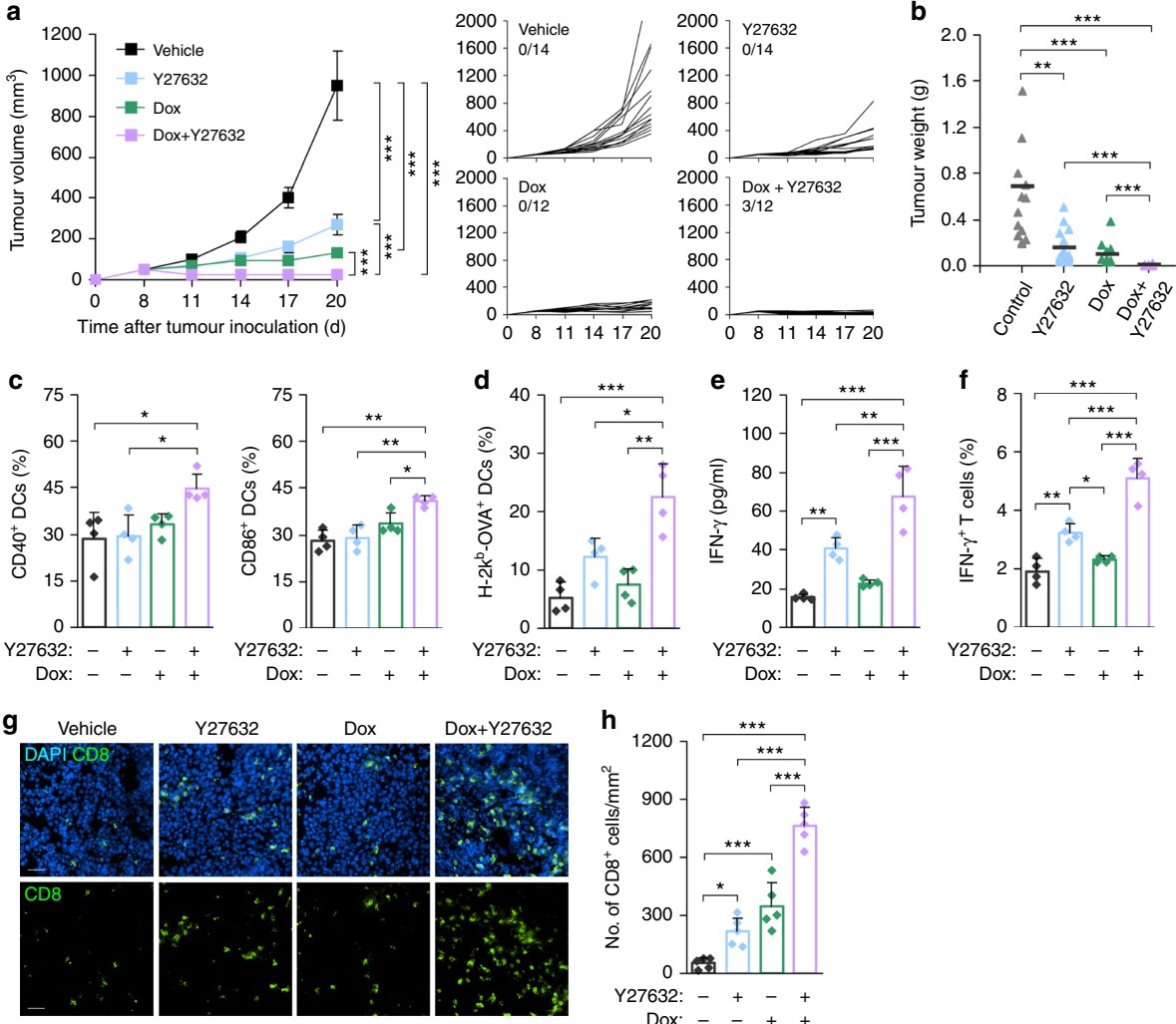

**Fig. 6** Combined therapy effectively induces antitumour immunity. **a**, **b** B16F10-Ova tumour-bearing C57BL/6 mice were injected (i.v.) with Y27632 (10 mg kg$^{-1}$) and/or Dox (5 mg kg$^{-1}$), as indicated in Supplementary Fig. 19. **a** Tumour volume was measured at the indicated times. Data are presented as means ± s.e.m. (left panel) and tumour growth curves of individual mice (right panels). Fractions indicate the number of mice showing complete tumour regression. **b** Tumour weights in individual mice were analysed at the end of the experiment. Bars indicate mean values ($n = 12$–14). **c** The percentage of CD40$^+$ and CD86$^+$ DCs among tumour-draining LN cells from B16F10-Ova tumour-bearing C57BL/6 mice treated Y27632 and/or Dox was analysed by flow cytometry. Data are presented as means ± s.d. ($n = 4$). **d** Tumour-draining LN cells were isolated from B16F10-Ova tumour-bearing C57BL/6 mice treated with Y27632 or vehicle, and the percentage of H-2k$^b$-OVA$^+$ DCs was analysed by flow cytometry using an antibody recognizing a complex of the ovalbumin peptide (SIINFEKL) with H-2k$^b$ (MHC-I). Data are presented as means ± s.d. ($n = 4$). **e** CD11c$^+$ DCs were isolated from tumour-draining LNs from B16F10-Ova tumour-bearing C57BL/6 mice treated with Y27632 and/or Dox and co-cultured with OT-I T cells for 72 h; secreted IFN-γ was analysed by ELISA. Data are presented as means ± s.d. ($n = 4$). **f** CD11c$^+$ DCs were isolated from tumour-draining LNs from B16F10-Ova tumour-bearing C57BL/6 mice treated with Y27632 and/or Dox and co-cultured with OT-I T cells. The percentage of IFN-γ-producing CD8$^+$ T cells was analysed by flow cytometry. Data are presented as means ± s.d. ($n = 4$). **g**, **h** B16F10-Ova tumour-bearing C57BL/6 mice were injected (i.v.) with Y27632 and/or Dox. On day 20, tumours were isolated and CD8$^+$ T cells in tumours were stained with anti-CD8 antibody. **g** A representative result. Scale bar, 50 μm. **h** The number of CD8$^+$ T cells per mm$^2$. Data are presented as means ± s.d. ($n = 5$). *$P < 0.05$, **$P < 0.01$, ***$P < 0.001$; significance determined by Kruskal–Wallis test with Bonferroni correction (**a**, **b**), or one-way ANOVA with Tukey's post hoc test (**c**–**f**, **h**)

immunostaining for CD8-positive cells in tumour tissues on 5 days after a course of treatment. Number of tumour-infiltrated CD8$^+$ T cells was remarkably increased by combined therapy compared with Y27632 or Dox treatment alone (Fig. 6g, h). Furthermore, to assess the therapeutic effect of combination of Y27632 and a non-ICD inducer in tumour-bearing mice, we determined the therapeutic dose of cisplatin that exerted a similar antitumour effect with 5 mg kg$^{-1}$ Dox in nude mice and administered B16F10-Ova tumour-bearing C57BL/6 mice with Y27632 alone or in combination with cisplatin. We found that the combination of cisplatin and Y27632 did not exert a more tumour-suppressing effect than Y27632 alone. Any additive effect

on the maturation and cross-presentation of DC was not found in combination of Y27632 and cisplatin (Supplementary Fig. 21). These results indicate that the combination of Y27632 and Dox effectively elicits an immune response against tumour antigen, thereby suppressing tumour growth.

**Combined therapy is effective in a genetic tumour model.** Autochthonous tumours from genetically engineered animals generate immunosuppressive tumour environments and are thus generally refractory to cancer immunotherapy[35]. To determine the efficacy of combined therapy in an autochthonous tumour

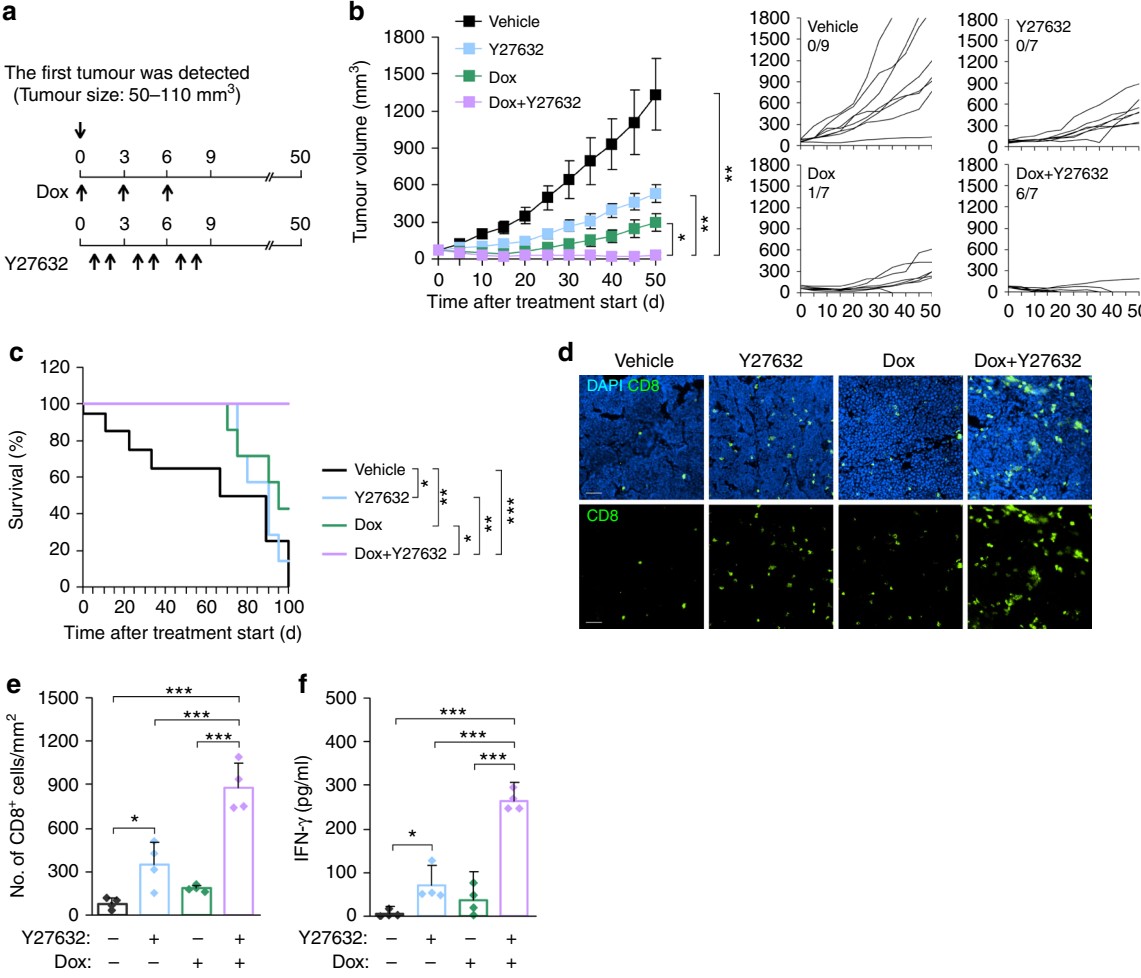

**Fig. 7** Combined therapy is effective in a genetic tumour model. **a** Schematic diagram of experiments for investigating the combined effect of Y27632 and Dox on tumour growth in a genetically engineered tumour model. **b** Tumour-bearing MMTV/Neu mice were injected (i.v.) with Y27632 (10 mg kg$^{-1}$) and/or Dox (5 mg kg$^{-1}$) as indicated in **a**. Tumour size was measured at the indicated times. Data are presented as means ± s.e.m ($n = 7$–9) (left panel) and tumour growth curves of individual mice (right panels). Fractions indicate the number of mice showing complete tumour regression. **c** Survival of tumour-bearing MMTV/Neu mice following combined therapy ($n = 7$–9). **d, e** MMTV/Neu mice bearing tumours were injected (i.v.) with Y27632 and/or Dox. On day 5 after Dox and/or Y27632 treatment, CD8$^+$ T cells in tumours were stained with an anti-CD8 antibody. **d** Representative result. Scale bar, 50 μm. **e** Number of CD8$^+$ T cells per mm$^2$. Data are presented as means ± s.d. ($n = 4$). **f** On day 5 after Dox and/or Y27632 treatment, single-cell suspensions from spleen were stimulated with rat Neu-derived peptide for 48 h, and IFN-γ production was measured by ELISA. Data are presented as means ± s.d. ($n = 4$). *$P < 0.05$, **$P < 0.01$, ***$P < 0.001$; significance determined by Kruskal–Wallis test with Bonferroni correction (**b**), log-rank test (**c**), or one-way ANOVA with Tukey's post hoc test (**e, f**)

model, we used the FVB/N-Tg (MMTVneu) 202Mul/J breast cancer model, in which the Neu oncogene is expressed under transcriptional control of the mouse mammary tumour virus (MMTV) promoter[36]. When the first palpable tumour (~50–110 mm$^3$) was identified, mice were injected (i.v.) with Dox alone or in combination with Y27632 (Fig. 7a). In vehicle-treated mice, tumours grew progressively and reached a size >1300 mm$^3$ 50 days after the start of treatment. Treatment with Dox or Y27632 alone reduced the tumour burden, but complete regression of tumour was not detected in any of the treated mice (Fig. 7b). In contrast, combined therapy with Dox and Y27632 efficiently promoted regression of induced tumours, resulting in complete regression in 85% (6 of 7) of mice (Fig. 7b) and improved overall survival (Fig. 7c). On the fifth day after a course of treatment, the infiltration of CD8$^+$ T cells into tumours was markedly increased in the group that received combined therapy (Fig. 7d, e). To assess whether combined therapy promotes an immune response against tumour-associated antigens, we isolated splenocytes from tumour-bearing MMTV/Neu mice and analysed

their immune response against Neu-derived peptide. IFN-γ production was markedly increased in mice treated with combined therapy compared with that in other groups (Fig. 7f). Collectively, these results clearly indicate that the combination of Y27632 and Dox provides therapeutic benefits in terms of enhanced tumour suppression, even in a genetic MMTV/Neu tumour model.

## Discussion

Blocking ROCK activity as a cancer therapy strategy has been extensively explored, with most attention having focused on tumour cell invasion and metastasis[6–9]. In this study, we provide multiple lines of evidence indicating that ROCK blockade modulates the function of phagocytes, enhancing tumour cell phagocytosis and thereby promoting T cell priming. First, we found that inhibition of RhoA/ROCK/myosin II pathway promoted the phagocytosis of cancer cells in vitro and in vivo. Second, the therapeutic effect of ROCK blockade was dependent on phagocytes and CD8$^+$ T cells. Third, ROCK blockade induced

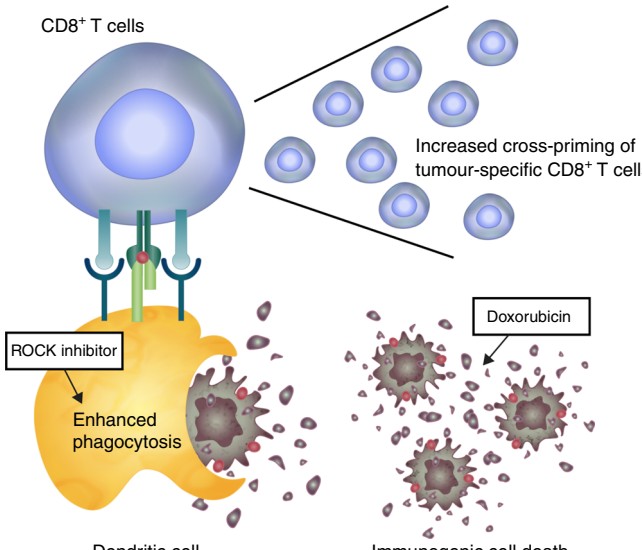

**Fig. 8** Combined therapy triggers and propagates immunity against cancer. Enhancing the phagocytic ability of APCs towards cancer cells undergoing immunogenic cell death increases cross-priming of tumour-specific CD8+ T cells, leading to T cell immunity against tumour cells

antitumour immunity by increasing the ability of DCs to cross-prime activation of CD8+ T cells. Furthermore, we here propose a therapeutic strategy combining ROCK blockade with chemotherapeutics that induce ICD of cancer cells. Our results demonstrated that the combination of ROCK blockade and an ICD inducer effectively promoted regression of tumour growth and triggered antitumour immunity in syngeneic and genetically engineered tumour models. The rationale for our combination therapy is that enhancing the phagocytic ability of APCs towards cancer cells undergoing immunogenic death potentiates T cell immunity against tumour cells (Fig. 8).

Uptake of tumour cells or tumour-specific antigens is the first stage of cancer immunity cycle to induce tumour-specific immune responses[29]. We found that ROCK inhibition increased cancer cell phagocytosis and immune responses for tumour antigen in a syngeneic tumour model. Cell depletion analyses revealed that phagocytes and cytotoxic T cells are important for therapeutic effect of ROCK blockade. We found that ROCK inhibition increased ability of DCs, but not macrophages, to prime T cell activation. However, although not statistically significant, phagocyte depletion by clodronate liposome partially suppressed tumour growth. Given that tumour-associated macrophages can play a protumoural role through different mechanisms[37], it is possible that existence of inhibitory macrophages masks the positive effect of Y27632 on macrophages. Cross-presentation of tumour antigens by DCs is an important mechanism for eliciting cytotoxic T cell responses for antitumour immunity[38]. Specific subtypes of DCs, such as migratory CD103+ DCs, promote trafficking of tumour-specific antigen, leading to priming antitumour T cell immunity[31,32]. Interestingly, we found that CD103+ DCs were increased in tumour-draining LNs and tumours from Y27632-treated mice. Cross-presentation of tumour antigen was markedly increased in CD103+ DCs from tumour-draining LNs and tumours of Y27632-treated mice. Recent studies showed CD103+ DCs was the main intratumoural myeloid population that transports tumour-specific antigens to the tumour-draining LNs[31]. In agreement with this finding, we found that tumour cell phagocytosis and DC maturation was increased in CD103+ DCs from tumours of Y27632-treated mice. Thus, our results suggest that

Y27632 enhanced the ability of DCs to prime T cell activation, possibly owing to an increase in phagocytosis of cancer cell-derived antigens by CD103+ DCs. On the other hand, it is possible that ROCK blockade directly or indirectly modulates the function of DCs. The effect of RhoA-ROCK blockade on DC function, however, remains controversial. Some studies have shown that ROCK inhibition impairs DC morphology and function[39,40], whereas others have reported that pharmacological inhibition of ROCK enhances antigen presentation by CD1d, an MHC class I-like molecule[41,42]. Moreover, a deficiency of a negative regulator of the RhoA/ROCK pathway was found to impair DC-mediated activation of T cells[43]. Furthermore, DC migration to draining LNs is important for the induction of adaptive immunity. Although activities of small GTPase Rac1 and cdc42 are crucial for cell migration, several studies showed that RhoA/ROCK blockade has a negative effect on DC migration velocity[40,44]. However, we found that CD103+ DC increased in tumour-draining LNs from Y27632-treated mice. A previous study showed that the reduction of actomyosin contractility via downregulation of RhoA activity and its downstream MLC phosphorylation contributed to CLEC-2-mediated DC migration along stromal cells scaffolds[45]. Thus, migratory CD103+ DCs might preferentially use a specific mechanism that causes efficient migration into LNs such as CLEC-2 signalling. Further dissection of the detailed molecular mechanism by which ROCK blockade unleashes DC-mediated T cell priming will be important for exploiting ROCK inhibitors for cancer immunotherapy.

ICD inducers can lead to increased expression of calreticulin, a prophagocytic signal for the uptake of tumour cells by DCs, on the surface of tumour cells and cause the release of ATP and high-mobility group box-1 (HMGB1) to promote DC maturation and cross-presentation, thereby promoting antitumour immunity[20,33,34]. Accordingly, we hypothesize that combination of ICD inducer and ROCK blockade potentiate DC-mediated phagocytosis and processing of cancer cells. Indeed, we found that ROCK blockade markedly enhanced DC-mediated phagocytosis of cell dying an immunogenic death in response to Dox. However, we found that Dox monotherapy did not significantly increase DC maturation or cross-presentation in vivo and exerted only a modest effect on T cell priming and infiltration in tumour-bearing mice. A previous study showed that immunogenic chemotherapy alone may be not sufficient to induce effective anti-tumour immunity in immunosuppressive tumour microenvironments[22], suggesting that further intervention is necessary to effectively promote antitumour immunity. Notably, combining Dox therapy with ROCK blockade augmented DC maturation and T cell priming in a syngeneic tumour model. This augmentation was not found in combination of Y27632 and a non-ICD inducer (cisplatin), suggesting that ICD of tumour cells by Dox contributes to potentiation of Y27632-mediated anti-tumour immunity. Moreover, increased expression of costimulatory markers was detectable in total CD11c+ DCs from tumour-draining LNs by only combination of Y27632 and Dox. It is unlikely that increased maturation of CD103+ DC population by Y27632 will be reflected in the total DCs. This might be explained in part by synergistic effect of enhanced uptake of tumour-associated antigen by DCs and activated pattern recognition receptor signalling by ICD-related damage-associated molecular patterns (e.g. HMGB1); however, the precise mechanism by which combined therapy enhances DC maturation remains to be clarified. A possible explanation for the effect of combined therapy is that both ICD induction and ROCK blockade contribute to making immunosuppressive tumour microenvironments more immunogenic and also drive immunogenic phagocytosis to efficiently induce antitumour immunity. ROCK-

mediated intracellular contractility has been reported to drive tumourigenesis by increasing ECM remodelling and tissue stiffness[11,12]. Fibrotic and immunosuppressive tumour microenvironments were further found to be associated with FAK hyperactivation, which is dependent on ROCK activity[46]. Moreover, danger signals from cancer cells dying an immunogenic death also contribute to the generation of ideal conditions for the initiation of antigen-specific immune responses. Notably, we found that the combination of Dox and Y27632 markedly augmented CD8[+] T cell infiltration in tumours in both syngeneic and MMTV/Neu tumour models compared to each agent alone. Furthermore, phagocytosis of dying cells under homeostatic conditions may restrict immune responses and facilitate protumourigenic responses (tolerogenic phagocytosis), whereas cancer cells killed by an ICD inducer could drive immunogenic phagocytosis and subsequently elicit antitumour immune responses through their antigenicity[47]. In this regard, the combination of a ROCK inhibitor and ICD inducer would be an important strategy for improving the efficacy of immunogenic phagocytosis and thereby potentiating antitumour immunity in an immunosuppressive tumour microenvironment.

In summary, we demonstrated that ROCK blockade elicited anticancer immunity by enhancing cancer cell phagocytosis as well as DC-mediated T cell priming. Therefore, modulation of the RhoA/ROCK pathway could represent a novel therapeutic strategy for regulating the phagocytic capacity of APCs in antitumour therapy. The combination of immunogenic killing and enhanced DC-mediated phagocytosis to augment tumour-specific immunity—which evokes the concept of awakening antitumour immunity against immunosuppressive tumours—is thus a promising strategy for maximizing the therapeutic efficacy of cancer immunotherapy.

## Methods

**Animals.** Eight-week-old male BALB/c and C57BL/6 mice were purchased from Orient Bio. OT-I CD8[+] T cell receptor (TCR)-Tg male mice and FVB/N-Tg (MMTVneu) 202Mul/J female mice were purchased from Jackson Laboratory. Mice were bred and maintained under specific pathogen-free conditions at the Korea Institute of Science and Technology (KIST). All experiments were conducted using protocols approved by the Association for Assessment and Accreditation of Laboratory Animal Care at the KIST.

**Reagents.** Y27632 dihydrochloride was purchased from Abcam. Neutralizing anti-CD4 (clone GK1.5, BE0003-1) and anti-CD8 (clone 2.43, BE0061) antibodies and isotype-matched rat IgG2b control (anti-KLH, clone LTF-2, BE0090) were obtained from BioXcell. Phycoerythrin (PE)-anti-CD86 (clone GL-1, 105008, 1:20), PE-anti-CD40 (clone 3-23, 124610, 1:20), allophycocyanin-anti-CD11b (clone M1/70, 101212, 1:100), allophycocyanin-anti-CD8α (clone 53-6.7, 100712, 1:100), allophycocyanin-anti-CD11c (clone N418, 117310, 1:100), allophycocyanin-anti-CD103 (clone 2E7, 121414, 1:100) and allophycocyanin-anti-F4/80 (clone BM8, 123116, 1:100) antibodies were from BioLegend. PE-mouse IgG1 (κ-isotype, MOPC-21, 400111, 1:200), PE-rat IgG1 (κ-isotype, RTK2071, 400407, 1:100), allophycocyanin-armenian hamster IgG (clone HTK888, 400912, 1:100), allophycocyanin-rat IgG2a (κ-isotype, clone RTK2758, 400512, 1:100), PE-rat IgG2a (κ-isotype, clone RTK2758, 400508, 1:20) and allophycocyanin-rat IgG2b (κ-isotype, clone RTK4530, 400612, 1:100) control were from BioLegend. Anti-CD16/CD32 (Fc blocker, clone 2.4G2, 553142, 1:50), anti-CD4 (clone RM4-5, 550280, 1:200) and anti-CD8α antibodies (clone 53-6.7, 550281, 1:200) and rat IgG2a (κ-isotype, R35-95, 559073, 1:1600) were obtained from BD Pharmingen. Clodronate liposomes and control liposomes were purchased from FormuMax Scientific Inc. Recombinant murine macrophage colony-stimulating factor (M-CSF) and TNF-related apoptosis-inducing ligand were obtained from Peprotech. Recombinant mouse Flt-3 ligand protein and IFN-γ ELISA Kit were purchased from R&D system. Dox hydrochloride, type IV collagenase, blebbistatin, cisplatin and type IV DNase I were obtained from Sigma. 5-Chloromethylfluorescein diacetate (CMFDA) and CellTracker Deep Red were purchased from ThermoFisher Scientific. pHrodo-SE, Alexa Fluor 488-Annexin V, propidium iodide and CFSE were obtained from Invitrogen. β-Gal$_{876–884}$ (TPHPARIGL), rat Neu$_{420–429}$ (PDSLRDLSVF)[48,49], P1A (LPYLGWLVF) and NP (RPQASGVYM) peptides were purchased from Peptron. F4/80, CD11c and CD8 MACS sorting systems were purchased from Miltenyi Biotech.

**Cell culture.** CT26.CL25 cells, which express high levels of both β-gal and the class I molecule H-2 Ld[28], were purchased from the ATCC and cultured in RPMI-1640 medium supplemented with 10% (v/v) foetal bovine serum (FBS; Invitrogen) and appropriate antibiotics. B16F10-Ova cells, which express high levels of chicken OVA, were provided by Dr. Seung-Hyo Lee (Korea Advanced Institute of Science and Technology, Daejeon, Republic of Korea) and maintained in Dulbecco's modified Eagle's medium (4.5 g L$^{-1}$ glucose) supplemented with 10% FBS and appropriate antibiotics. BMDMs were generated by isolating bone marrow cells from 8-week-old male BALB/c or C57BL/6 mice and culturing them for 7 days in the presence of 20 ng ml$^{-1}$ M-CSF (Peprotech)[50]. For differentiation into BMDCs, bone marrow cells were incubated with 200 ng ml$^{-1}$ Flt-3 ligand (R&D Systems) and 0.1% β-mercaptoethanol for 10 days. CD8[+] OT-I T cells were isolated from 8-week-old male naïve OT-I mice using a mouse CD8[+] T cell enrichment column (R&D Systems). A tumour cell line expressing mCherry was generated by transducing CT26.CL25 cells with retrovirus particles encoding mCherry. After 24 h, the cells were cultured with RPMI-1640 medium containing 10 μg ml$^{-1}$ puromycin for 2 weeks. mCherry expression was verified by flow cytometry. All cell lines used in this study were tested for mycoplasma contamination.

**Phagocytosis assay.** For flow cytometry analyses, BMDMs or BMDCs were stained with CellTracker Deep Red (1 μM; ThermoFisher Scientific) for 30 min at 37 °C and plated at a density of $2 \times 10^5$ cells per 35 mm dish. The next day, CT26.CL25 or B16F10-Ova cells were stained with CMFDA (0.5 μM; ThermoFisher Scientific) and co-cultured with syngeneic BMDMs or BMDCs at a ratio of 1:1 in the presence or absence of Y27632 (30 μM; Abcam) or blebbistatin (5–20 μM; Sigma). After incubation for 2 h at 37 °C, the phagocytosis of cancer cells was analysed by flow cytometry (Accuri C6; BD Biosciences) using the FlowJo (v10) software. Phagocytosis (%) was calculated according to the fomula, number of phagocytes harbouring cancer cells (right-upper quadrant)/total number of phagocytes (right quadrants) × 100. In some experiments, at 0, 0.5, 1, 2 and 4 h after Y27632 washout, CT26.CL25 or B16F10-Ova cells were added into BMDMs or BMDCs and co-cultured for 2 h, and phagocytosis (%) was analysed by flow cytometry. For phagocytosis of Dox-treated cancer cells, CT26.CL25 or B16F10-Ova cells were treated with 25 or 2.5 μM Dox, respectively, for 24 h at 37 °C to induce cell death. CellTracker Deep Red-labelled BMDMs or BMDCs were incubated with CMFDA-stained, Dox-treated cancer cells for 5 min at 37 °C, and phagocytosis (%) was determined by flow cytometry. In some experiments, BMDMs, cancer cells, or both were treated with Y27632. For phagocytosis of tumour cells dying a non-immunogenic and necrotic cell death, CT26.CL25 or B16F10-Ova cells were treated with cisplatin (150 μM, 24 h) or incubated at 55 °C for 30 min to induce non-ICD or necrosis, respectively. For analysis of phagocytic activity in DC subtypes, CD11c[+] DCs were isolated from tumours in B16F10-Ova tumour-bearing mice treated with Y27632 or vehicle and incubated with CMFDA-stained B16F10-Ova cells for 2 h. The phagocytosis in CD11b[+] and CD103[+] subtypes was analysed by flow cytometry using allophycocyanin-conjugated anti-CD11b (clone M1/70, 101212. 1:100) or anti-CD103 antibody (clone 2E7, 121414, 1:100).

For microscopic analyses, phagocytosis assays were performed using the pH-sensitive pHrodo dye[23]. Briefly, CMFDA-stained BMDMs or BMDCs were pretreated with Y27632 (30 μM) or vehicle for 1 h and then incubated with pHrodo-SE-labelled CT26.CL25 cells for 2 h at 37 °C in the presence of Y27632 (30 μM). The cells were washed with basic PBS (pH 10.0) to quench the fluorescence of un-engulfed, pHrodo-SE-labelled cancer cells. Cancer cell phagocytosis was then imaged under an inverted fluorescence microscope (Nikon), and phagocytosis (%) was determined in seven or more randomly selected microscopic fields per experiment.

For analysis of phagocytic activity in macrophages and DCs from mice that are not bearing tumour, CFSE-stained apoptotic thymocytes ($1 \times 10^7$) were injected (i.v.) into the 8-week-old C57BL/6 mice treated with Y27632 (10 mg kg$^{-1}$) as indicated in Supplementary Fig. 9a. After 1 h, F4/80[+] macrophages and CD11c[+] DCs were isolated from splenocytes and phagocytosis (%) was analysed by flow cytometry.

**In vivo tumour models.** Subcutaneous syngeneic tumour models were generated by subcutaneously (s.c.) injecting the right flank of 8-week-old male BALB/c mice or C57BL/6 mice with $1 \times 10^6$ CT26.CL25 cells or $5 \times 10^5$ B16F10-Ova cells. When tumour size reached ~50–100 mm$^3$, Y27632 (10 mg kg$^{-1}$) or vehicle (PBS) was injected (i.v.) as indicated in Supplementary Fig. 6a, b. Tumours were measured every 3 days, and volume was calculated according to the formula, (width$^2$ × length)/2. For nude mice, $1 \times 10^6$ CT26.CL25 cells were injected into the right flank of 8-week-old male BALB/c nude mice, and 10 mg kg$^{-1}$ of Y27632 was injected (i.v.) as indicated in Supplementary Fig. 6d. For antitumour vaccination experiments, primary CT26.CL25 tumours were surgically removed 22 days after tumour inoculation. Seven day later, $7 \times 10^6$ CT26.CL25 cells were injected into the contralateral flank of BALB/c mice, and tumour growth was monitored for 25 days as indicated in Supplementary Fig. 6f. For combination therapy, B16F10-Ova tumour-bearing mice were treated with Y27632 (10 mg kg$^{-1}$, i.v.) alone or in combination with Dox (5 mg kg$^{-1}$, i.v.) or cisplatin (3 mg kg$^{-1}$, i.v.) as indicated in Supplementary Fig. 18. For the MMTV/Neu tumour model, female mice were monitored for tumour development by palpation every 5 days. When tumours reached a size

of ~50–110 mm$^3$ (26–32 weeks of age), mice were administered Y27632 (10 mg kg$^{-1}$) alone or in combination with Dox (5 mg kg$^{-1}$) as indicated in Fig. 7a. Tumour volume was determined every 3–5 days, and mice were euthanized when tumour volume exceeded 2500 mm$^3$ or if mice exhibited signs of impaired health. The experiments were not randomized and the investigators were not blinded to allocation during experiments and outcome assessment.

**Cell depletion experiments**. For phagocyte depletion, 200 μl of clodronate liposomes (FormuMax Scientific Inc.) was initially injected (i.p.) 1 day before Y27632 treatment, followed by injection of 100 μl of clodronate liposomes every 4 days, as indicated in Supplementary Fig. 6c. PBS liposomes (FormuMax Scientific Inc.) were used as a negative control. Depletion of F4/80$^+$ macrophages or CD11c$^+$ DCs was verified by flow cytometry analysis using an allophycocyanin-conjugated anti-F4/80 (clone BM8, 123116, 1:100) or anti-CD11c (clone N418, 117310, 1:100) antibody, respectively. For CD4$^+$ or CD8$^+$ T cell depletion, 200 μg neutralizing anti-CD4 (clone GK1.5) or anti-CD8 antibody (clone 2.43) was injected 1 day before Y27632 treatment and then every 3 days thereafter, as indicated in Supplementary Fig. 6e. Rat IgG2b (clone LTF-2) was used as an isotype-matched control. Depletion of CD4$^+$ or CD8$^+$ cells was verified by flow cytometry using an anti-CD4 (clone RM4-5, 550280, 1:200) or anti-CD8 (clone 53-6.7, 100712, 1:100) antibody, respectively.

**Flow cytometry**. Tumours and tumour-draining LNs were isolated from tumour-bearing mice. Tumour-draining LNs were mechanically digested, and tumours were digested with DNase I and collagenase. Single-cell suspensions were treated Fc blocker (CD16/CD32, clone 2.4G2, 553142, 1:50) for 5 min at 4 °C and then stained with allophycocyanin-conjugated anti-F4/80 (clone BM8, 123116, 1:100), allophycocyanin-conjugated anti-CD11c (clone N418, 117310, 1:100), PE-conjugated anti-CD40 (clone 3.23, 124610, 1:20), PE-conjugated anti-CD86 (clone GL-1, 105008, 1:20), or PE-conjugated anti-IFN-γ (clone XMG1.2, 505807, 1:100) antibody for 1 h at 4 °C. After wash with PBS twice times, the cells were subjected to flow cytometry. For analysis of IFN-γ-positive cells, cells were fixed with 1% paraformaldehyde for 7 min and permeabilized with 0.1% Triton X-100 for 3 min before the addition of antibody. For analysis of PS exposure on the cell surface, CT26.CL25 or B16F10-Ova cells were treated with 30 μM Y27632 for 24 h, stained with Alexa Flour 488-labelled Annexin V, and analysed by flow cytometry.

For DC subset analysis, CD11c$^+$ DCs were isolated from tumour-draining LNs and tumours of B16F10-Ova tumour-bearing mice treated with Y27632 or vehicle through CD11c MACS sorting system. The percentages of CD11b$^+$, CD103$^+$ and CD8$^+$ DCs were analysed by flow cytometry using allophycocyanin-conjugated anti-CD11b (M1/70, 101212, 1:100), anti-CD103 (2E7, 121414, 1;100), or anti-CD8 (53-6.7, 100712, 1:100) antibodies, respectively. Allophycocyanin-conjugated rat IgG2b (κ-isotype, clone RTK4530, 400612, 1:100), rat IgG2a (κ-isotype, clone RTK2758, 400512, 1:100) and Armenian hamster IgG (clone HTK888, 400912, 1:100) were used as the isotype controls for anti-CD11b, anti-CD8 and anti-CD103 antibodies, respectively.

**IFN-γ measurement**. CT26.CL25-mCherry tumour-bearing mice were treated with Y27632 or vehicle. Twenty days after tumour inoculation, tumour-draining LNs were extracted from mice, and single-cell suspensions from tumour-draining LNs were incubated with 5 μg ml$^{-1}$ of a β-gal-derived peptide (TPHPARIGL); P1A peptide (LPYLGWLVF) was used as a negative control. After incubation for 48 h, the amount of IFN-γ in the culture medium was measured using a mouse IFN-γ Quantikine ELISA Kit (R&D Systems). Antigen-specific IFN-γ production was obtained by subtracting nonspecific production obtained using the negative control peptide. In some experiments, CD8$^+$ and CD8$^-$ cells were isolated from tumour-draining LN cells from tumour-bearing mice, and IFN-γ production in response to β-gal-derived peptide was analysed. For tumour-bearing MMTV/Neu mice, splenocytes were isolated 5 days after Y27632 and/or Dox treatment, and a single-cell suspension was incubated with 10 μg ml$^{-1}$ rat Neu-derived peptide (PDSLRDLSVF); NP peptide (RPQASGVYM) was used as a negative control.

**Analysis of DC maturation**. For in vitro DC maturation, CMFDA-stained BMDCs were pretreated with Y27632 (30 μM) or vehicle for 1 h and co-cultured with CT26.CL25 cells or Dox-treated CT26.CL25 cells for 4 h at 37 °C in the presence of Y27632 or vehicle. After removal of un-engulfed cells, BMDCs were incubated in the presence of Y27632 or vehicle for an additional 20 h. The percentage and mean fluorescence intensity (MFI) of CD40$^+$ or CD86$^+$ BMDCs was then analysed by flow cytometry using allophycocyanin-conjugated anti-CD40 (clone 3/23, 124611, 1:20) and anti-CD86 (clone GL-1, 105011, 1:100) antibodies; allophycocyanin-conjugated IgG2a (clone RTK2758, 400512, 1:100) was used as an isotype control. For in vivo analysis, B16F10-Ova tumour-bearing mice were administered Y27632 (10 mg kg$^{-1}$, i.v.) and/or Dox (5 mg kg$^{-1}$, i.v.). On day 20 after tumour inoculation, the percentage and MFI of CD40$^+$ or CD86$^+$ DCs among tumour-draining LN cells from tumour-bearing mice was analysed by flow cytometry using allophycocyanin-conjugated anti-CD11c (clone N418, 117310, 1:100) and PE-conjugated anti-CD40 (clone 3.23, 124610, 1:20) or anti-CD86 antibodies (clone GL-1, 105008, 1:20). In some experiments, CD11c$^+$ DCs were isolated from

tumour-draining LN, and the percentage and MFI of CD40$^+$ cells in CD11c$^+$CD103$^+$ DCs or CD11c$^+$CD8$^+$ DCs was analysed by flow cytometry using PE-conjugated anti-CD40 antibody and allophycocyanin-conjugated anti-CD103 (clone 2E7, 121414, 1:100) or anti-CD8 antibody (clone 53-6.7, 100712, 1:100).

**T cell priming assay**. Cross-priming of CD8$^+$ T cells was analysed by measuring the proliferation of OVA-specific T cells (OT-I) using a carboxyfluorescein succinimidyl ester (CFSE) dilution assay[51]. Briefly, B16F10-Ova tumour-bearing mice were treated (i.v.) with Y27632 and/or Dox; 20 days after tumour inoculation, 2 × 10$^6$ CFSE-labelled CD8$^+$ OT-I T cells were injected (i.v.) into mice. Three days later tumour-draining LNs were removed, and proliferation of CSFE-labelled OT-I T cells from tumour-draining LN cells was analysed by flow cytometry. For ex vivo T cell priming assays, B16F10-Ova tumour-bearing mice were treated (i.v.) with Y27632 and/or Dox. Tumours and tumour-draining LNs were extracted 20 days after tumour inoculation and enzymatically digested by incubating with collagenase IV and DNase I for 1 h. Cells from tumours and tumour-draining LNs were then mechanically dissociated through a 40-μm cell strainer, and DCs and macrophages were purified using CD11c and F4/80 MACS sorting systems (Miltenyi Biotech), respectively. DCs or macrophages (5 × 10$^4$ each) were co-cultured with 2.5 × 10$^5$ purified OT-I T cells for 3 days. The supernatants were collected, and IFN-γ was measured using a mouse IFN-γ Quantikine ELISA Kit. The cells were fixed with 4% paraformaldehyde and permeabilized with 0.2% Triton X-100, and the percentage of IFN-γ-producing CD8$^+$ T cells was analysed by flow cytometry using allophycocyanin-conjugated anti-CD8α (clone 53-6.7, 100712, 1:100) and PE-conjugated anti-IFN-γ (clone XMG1.2, 505807, 1:100) antibodies. PE-conjugated rat IgG1 (κ-isotype, RTK2071, 400407, 1:100) was used as the isotype control for anti-IFN-γ antibody.

**Cross-presentation analysis**. Detection of OVA peptide–MHC-I complexes on the surface of DCs was analysed by flow cytometry[34,52]. For in vitro analyses, CMFDA-stained BMDCs were pretreated with Y27632 or vehicle for 1 h and co-cultured with 2 × 10$^5$ B16F10-Ova cells or Dox-treated B16F10-Ova cells for 4 h. After removal of un-engulfed cells, BMDCs were incubated for an additional 20 h. The percentage of H-2k$^b$-OVA$^+$ BMDCs was analysed by flow cytometry using allophycocyanin-conjugated anti-H-2k$^b$-OVA (clone 25-D1.16, 141605, 1:20) antibody. For in vivo analyses, B16F10-Ova tumour-bearing mice were treated (i.v.) with Y27632 and/or Dox; 4 days after treatment, the percentage of H-2k$^b$-OVA$^+$ DCs among tumour-draining LN cells was analysed by flow cytometry using allophycocyanin-conjugated anti-CD11c (clone N418, 117310, 1:100) and PE-conjugated anti-H-2k$^b$-OVA (clone 25-D1.16, 141603, 1:200) antibodies. For analysis in DC subsets, CD11c or DCs were isolated from tumour-draining LNs and tumours, and the percentages of H-2k$^b$-OVA$^+$ cells in CD11b$^+$, CD103$^+$ or CD8$^+$ DCs were analysed by flow cytometry using PE-conjugated anti-H-2k$^b$-OVA antibody and allophycocyanin-conjugated anti-CD11b (M1/70, 101212, 1:100), anti-CD103 (clone 2E7, 121414, 1:100), or anti-CD8 antibody (clone 53-6.7, 100712, 1:100). PE-conjugated mouse IgG1 (κ isotype, MOPC-21, 400111, 1:200) was used as the isotype control for anti-H-2k$^b$-OVA antibody.

**Immunostaining**. Tumour tissues were dehydrated and embedded in OCT compound in cryomolds, then cut into 10-μm sections using a rotary microtome. After blocking nonspecific immunoglobulin binding by incubating with PBS supplemented with 3% bovine serum albumin for 1 h, sections were incubated with anti-CD8α antibody (clone 53-6.7, 550281, 1:200) at 4 °C overnight, washed extensively with PBS, and then incubated with Alexa Fluor 488-conjugated secondary antibody (Jackson ImmunoResearch, 712-546-153, 1:400). Purified rat IgG2a (κ-isotype, clone R35-95, 559073, 1:1,600) was used as isotype-matched control. CD8$^+$ cells were visualized by fluorescence microscopy (Nikon), and the number of CD8$^+$ T cells per mm$^2$ was calculated using the Image J program.

**Statistical analysis**. All statistical analyses were carried out using the SPSS software (v23). Sample sizes in experiments were chosen on the basis of sizes reported in this field. Data were first analysed for a normal distribution using a Shapiro–Wilks test. Comparisons between two groups were performed using unpaired two-tailed Student's $t$ test (for normally distributed data) or Mann–Whitney test (for non-normally distributed data). One-way analysis of variance (ANOVA) was used for comparisons of more than two groups, and multiple comparisons were performed using a Tukey's post hoc test. In cases where data were not normally distributed, a Kruskal–Wallis test was used, and multiple comparisons were performed using a Mann–Whitney test with Bonferroni correction. Statistical significance for tumour-free frequency and survival benefit was determined using a log-rank (Mantel–Cox) test. All results are expressed as means ± s.d. or means ± s.e.m. $P$ values <0.05 were considered statistically significant; individual $P$ values are indicated in figure legends.

**Data availability**. All relevant data are available with the article and its Supplementary Information files, or available from the corresponding authors upon reasonable request.

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

## Acknowledgements

This work was supported by a grant from the National Research Foundation (NRF) of Korea funded by the Korean government (2017R1A3B1023418), the National R&D Program for Cancer Control, Ministry of Health and Welfare, Republic of Korea (1420390), the KU-KIST Graduate School of Converging Science and Technology Program and the KIST Institutional Program.

## Author contributions

I-.S.K. and S-.Y.P. designed the project and experiments. I-.S.K., S-.Y.P. and G-.H.N. analysed all data. I-.S.K. and S-.Y.P. wrote the manuscript. G-.H.N., E.J.L., Y.K.K., Y.H., Y.C. and M-.J.R. performed in vitro experiments. G-.H.N. and E.J.L. carried out in vivo

experiments. I-.C.K., Y.Y. and D.J.A. provided advice and made suggestions for the manuscript. J.W. and Y.C. contributed to animal management.

## Additional information

**Competing interests:** The authors declare no competing interests.

