## [Peer Review File · Nature Communications]

Reviewers' comments:

Reviewer #1 (Remarks to the Author):

In this work, Nam et al. have demonstrated that inhibition of the kinase activity of ROCK by Y27632 promotes macrophages and DCs phagocytosis both in vitro and in vivo. Importantly, the authors reported that Y27632 treatment has significant therapeutic effect on cancer in mice model. In addition, they dissected the underlying mechanism and unraveled that the therapeutic effect of Y27632 is dependent on DCs. The authors also showed that Y27632 in combination with ICD inducer "doxorubicin" treatment had even stronger therapeutics effect. Based on these findings, the authors suggested that a therapeutic strategy that combines enhanced phagocytosis and immunogenic killing led to synergistic CD8+ cytotoxic T-cell priming and infiltration into tumor, and thus is promising for cancer immunotherapy.

Given that the related Rho kinase inhibitors such as fasudil and the anthracyclines chemotherapy agents are approved drugs available in clinic worldwide, Nam et al. findings are likely to have immediate implication for cancer treatment in human patients. Moreover, their discovery of a novel function of ROCK in DCs is potentially interesting. Therefore, if validated solidly by experiments, I believe that this work will have strong and broad impact in both the field of cell biology of ROCK and cancer immunology.

However, it should be noted that the concept that ROCK kinase activity plays a role in phagocytosis was previously reported by the authors' group (Kim et al., Plos ONE 12, e0174603, 2017) and others (Tosello-Tramont et al., JBC 278, 49911-49919, 2003). Therefore, it is difficult to evaluate this finding as a significant conceptual advance unless the authors can address some aspects of the underlying mechanistic basis, e.g. the involvement of actin cytoskeleton. Moreover, although the authors showed a correlation between increased phagocytosis of DCs and the therapeutic effect of Y27632 treatment, their causal-relationship is unclear. As this issue is critical to explain why Y27632 could exert its therapeutic effect in cancer model, it is necessary to address how increased phagocytosis by DCs actually promotes cancer immunity. The authors should also beware that accumulating evidences suggested that specific subtypes of DCs (e.g. CD103+ cDCs) promotes cancer immunity while other subtypes of DCs induce tolerance (Veglia and Gabrilovich, Curr Opin Immun 45, 43-51, 2017). Thus, increase of antigen presentation by DCs may not simply facilitate the immunogenicity to cancer, if that occurs in the DCs subtypes promoting tolerance. Additional analyses about the DCs subtypes are needed because they may dramatically change the interpretation of the current data set. Finally, although combination of Y27632 and doxorubicin had obviously strong therapeutic effect in cancer model, their effects appeared to be additive rather than synergistic. Investigation of the effect of combination therapy of Y27632 with other cancer drugs of different mechanisms, e.g. checkpoint inhibitor anti-PD1 Ab, maybe helpful to further clarify this issue.

Major comments

1) The authors concluded from Fig 1h and Fig S3 that ROCK blockade augments phagocytosis by modulating phagocytes rather than by acting on tumor cells, because only the pre-treatment of Y27632 with phagocytes but not tumor promotes the phagocytosis ability. Given that Y27632 is a reversible inhibitor (Narumiya et al., Methods Enzymol 325, 273-284, 2000) and rapidly recover upon washout in vitro within few minutes (for example, see Nakazawa et al., PNAS 113, E6813-E6822, 2016), it is surprising that pre-treatment of phagocytes with Y27632 for 2 hours could strongly suppress the phagocytosis activity. Had the authors ever checked that the effect of pretreatment of Y27632 on phagocytes for 2 hours long-lasting or reversible? If the effect is long-lasting, there is a possibility that Y27632 change the phagocytes characteristic (e.g. cell differentiation) critical for their phagocytosis ability rather than directly regulate the phagocytosis process. It is necessary to clarify this important issue.

2) The authors showed in Fig 4 that although there was no significant change in the maturation of DCs, Y27632 treated-CD11c+ DCs isolated from tumor-draining LNs and tumor significantly promotes antigen-specific T cell activation as evident by IFN gamma production. It is clear that T-cell priming by DCs is important for eliciting an antitumor immune response after Y27632 therapy but what is the mechanistic explanation for this Y27632-treated DCs T-cell priming ability? Is there a direct causal-relationship between DCs T cell priming ability and the increased phagocytosis activity of DCs they reported in Fig 1 and Fig 2? In addition, analyses of the DCs subtypes in tumor-draining LNs and tumor are needed because the changes affect antitumor immune response. Did the authors observe any increase or decrease of particular DCs subtypes in their model upon Y27632 treatment?

3) In Fig 6, the authors obviously showed that combination of Y27632 and an ICD inducer efficiently suppresses tumor growth. However, although only the combination of Y27632 and Dox could significantly induce DCs maturation as demonstrated in Fig 6c & 6d and be considered as synergistic effect, its importance on antitumor immunity is unclear because the effect on 3 important parameters of antitumor immunity including, the reduction of tumor size, percentage of H-2Kb-OVA+ DCs and IFN gamma production by antigen-specific OT-I T cells, look additive rather than synergistic. It may be helpful to see the therapeutic effect of the combination of Y27632 with other anti-cancer drugs, such as anti-PD1 antibody and compare the results with the current findings.

Minor comments

1) Fig 2d. I'm not sure about the statistics significance, but clodronate treatment reduced the tumor size. This finding may suggest the existence of immune suppression macrophages in this model. Could the authors add a discussion about this issue in their manuscript?

2) Fig4a. The author presented their data as percentage of cells in tumor-draining LNs, but I'm wondering what about the absolute cell numbers. Although the percentage is not significantly different, is it possible that the absolute cell number is larger in the Y27632-treated group and therefore results in stronger antitumor immunity.

3) Fig 4d-e. The authors showed that Y27632 treatment of F4/80+ macrophages isolated and purified from TDLNs or tumors has no positive effect on T cell activation. However, as results in Fig 2d suggested the existence of inhibitory macrophages, it is possible that they are included in the purified F4/80+ macrophages used in this experiment and therefore mask the positive effect of Y-27632 in macrophages. Could the authors give some comments about this issue?

4) The results of Fig 5a together with Fig 1c suggested that Y27632 promotes BMDCs phagocytosis ability regardless of the nature of their target cells, live of Dox-treated ICD. Does the phagocytosis by Y27632-treated BMDCs has any selectivity against specific type of cells (or death cells)? Could Y27632-treatment also promote phagocytosis of target cells undergoing necrosis or non-ICD apoptosis?

5) Fig 5a. X axis should be corrected as BMDCs (CMFDA-DR).

6) In the introduction of this paper, the authors cited Liu X et al. and Sockolosky JT et al. works and mentioned that "therapeutic agents that antagonize CD47, which binds to the receptor, SIRPa (signal-regulatory protein alpha), on macrophages and simulates anti-phagocytic signaling, have been found to drive T cell mediated elimination of immunogenic tumours". According to these 2 works, it was shown that blockage of CD47 results in increased phagocytosis in both macrophages and DCs. In addition, it was also shown that CD47-specific blocking involved in the T cell mediated elimination of immunogenic tumors. However, according to the latest related-work (Xu et al., Immunity 47, 363-373, 2017), the underlying mechanism of anti-tumor effect by CD47 blocking is largely depended on the sensing of mtDNA in the DCs subtype producing type I interferon but not the increased phagocytosis ability of macrophages and DCs. The authors should also cite this

work.

7) What are the identity of the gray peaks in Fig 2e, Fig 5e, Supplementary Fig 1, Supplementary Fig 6, Supplementary Fig 7 and Supplementary Fig 8? I couldn't find explanations in the figure legends. Are they the staining with isotype control?

8) Several published works suggested the inhibitory role of Y-27632 on DC migration and therefore immunosuppression (e.g. Lammermann et al., *Nature* 453, 51-55, 2008; Nitschke et al., *Blood* 120, 2249-2258, 2012). How could the authors' findings in the present work reconcile with these previous findings? Could the authors provide some discussions on this issue?

Reviewer #2 (Remarks to the Author):

The manuscript by Nam et al., which is well-written, presents evidence for the synergistic antitumour activity of a chemotherapeutic (e.g., doxorubicin) and rho kinase (ROCK) inhibitor in implanted as well as spontaneous tumors. The initial focus is on ROCK inhibition as a stand-alone analysis. In vitro, Y27632 treatment of BMDM or BMDC cultured with either the CT26.CL25 (colon cancer) or B16F10-Ova (melanoma) tumour lines resulted in enhanced phagocytosis of the tumour cells. However, looking at the numbers that the authors provided regarding percent of phagocytosis, it was unclear exactly how it was calculated. It is mentioned in the Methods, but my calculations come out differently.

Most of the in vitro experiments that you would want to see were there, as were the appropriate in vivo challenge and spontaneous tumour model systems; the appropriate statistical analyses were performed. Unfortunately, particularly for the in vitro experiments, certain controls were not always included.

Although the data are certainly consistent with the authors premise, there are some holes that do not fully pin down the mechanism of antitumour activity. This has reduced this reviewer's overall enthusiasm for this manuscript.

Specific questions, comments and criticisms:

1. Supplemental Fig. 2 is meant to show that the ROCK inhibitor Y27632 does not induce apoptosis in the tumour cells. An important positive control would be treating the tumour cells with a drug that definitely causes activation-induced cell death.
2. In Fig. 2, macrophages and DC from Y27632-treated tumour-bearing mice were better at phagocytosing tumour cells than were those from vehicle-treated tumour-bearing mice. Does Y27632 itself enhance phagocytic activity in these cell types from mice that are not bearing tumours?
3. Which cells are making IFN-gamma in the draining LNs? How do you know?
4. In the experiment presented in Fig. 5g, the authors argue that "Cross-presentation of an OVA-derived peptide to MHC-I was markedly increased by co-culture of BMDCs with Dox-treated B16-F10-Ova cells in the presence of Y27632..." Actually, without Dox, there is a greater increase in cross-presentation (2.2-fold increase vs. 1.66-fold with Dox). This would suggest Dox does not enhance the cross-presentation of the Kb/OVA complex.
5. In Fig. 5e, co-culture of BMDCs with Dox-treated tumour cells had enhanced (albeit really only modestly) CD40 and CD86 surface expression. Fig. 4 looks at the percentage of BMDCs that are CD40+ and CD86+ and Y27632 treatment made no difference, but no analysis of surface expression by flow cytometry was apparently done. What effect does Y27632 treatment have on co-stimulatory molecule expression on BMDCs co-cultured with tumour cells not treated with Dox?
6. Fig. 7c shows that tumour-bearing mice treated with both Dox and Y27632 had 100% survival. This is remarkable. How many times was this experiment performed? Did you always get 100% survival with this dual treatment regimen?

Minor question:

1. Kb binds peptides that are 8 or 9 amino acids in length. The Neu-derived peptide is a 10-mer. Is this a common sequence for this molecule in FVB (H-2q) mice? It was not cited.

Response to Reviewers' comments:

Reviewer #1 (Remarks to the Author):

In this work, Nam et al. have demonstrated that inhibition of the kinase activity of ROCK by Y27632 promotes macrophages and DCs phagocytosis both in vitro and in vivo. Importantly, the authors reported that Y27632 treatment has significant therapeutic effect on cancer in mice model. In addition, they dissected the underlying mechanism and unraveled that the therapeutic effect of Y27632 is dependent on DCs. The authors also showed that Y27632 in combination with ICD inducer “doxorubicin” treatment had even stronger therapeutics effect. Based on these findings, the authors suggested that a therapeutic strategy that combines enhanced phagocytosis and immunogenic killing led to synergistic CD8+ cytotoxic T-cell priming and infiltration into tumor, and thus is promising for cancer immunotherapy.

Given that the related Rho kinase inhibitors such as fasudil and the anthracyclines chemotherapy agents are approved drugs available in clinic worldwide, Nam et al. findings are likely to have immediate implication for cancer treatment in human patients. Moreover, their discovery of a novel function of ROCK in DCs is potentially interesting. Therefore, if validated solidly by experiments, I believe that this work will have strong and broad impact in both the field of cell biology of ROCK and cancer immunology.

However, it should be noted that the concept that ROCK kinase activity plays a role in phagocytosis was previously reported by the authors' group (Kim et al., Plos ONE 12, e0174603, 2017) and others (Tosello-Tramont et al., JBC 278, 49911-49919, 2003). Therefore, it is difficult to evaluate this finding as a significant conceptual advance unless the authors can address some aspects of the underlying mechanistic basis, e.g. the involvement of actin cytoskeleton.

[Answer]

RhoA/ROCK pathway has been found to modulate the regulatory myosin light chain (MLC) by direct phosphorylation or by inhibiting MLC phosphatase, leading to actomyosin assembly and cell contraction. It is possible that decrease of contractility by ROCK blockade facilitates cell shape change for effective phagocytosis. To investigate the mechanism by which ROCK blockade enhances cancer cell phagocytosis, we analyzed the effect of inhibition of the motor protein myosin II on cancer cell phagocytosis. Treatment of BMDMs or BMDCs with a myosin II inhibitor, blebbistatin, led to a significant increase in the engulfment of CT26.CL25 and B16F10-Ova cells, reminiscent of that observed for phagocytes treated with Y27632. This result suggests that tumour cell phagocytosis is modulated by RhoA/ROCK/myosin II pathway

This result was incorporated into Fig. 1i,j and Supplementary Fig. 5.

The description for the result was added in Results section (p. 6, lines 6-14).

The experimental procedure was added in Methods section (p. 20, line 8).

In p.14, line 6, “treatment of Y27632” was changed into “inhibition of RhoA/ROCK/myosin II pathway”.

In p.14, line 7, “Y27632” was changed into “ROCK blockade”.

Moreover, although the authors showed a correlation between increased phagocytosis of DCs and the therapeutic effect of Y27632 treatment, their causal-relationship is unclear. As this issue is critical to explain why Y27632 could exert its therapeutic effect in cancer model, it is necessary to address how increased phagocytosis by DCs actually promotes cancer immunity. The authors should also beware that accumulating evidences suggested that specific subtypes of DCs (e.g. CD103+ cDCs) promotes cancer immunity while other subtypes of DCs induce tolerance (Veglia and Gabrilovich, Curr Opin Immun 45, 43-51, 2017). Thus, increase of antigen presentation by DCs may not simply facilitate the immunogenicity to cancer, if that occurs in the DCs subtypes promoting tolerance.

Additional analyses about the DCs subtypes are needed because they may dramatically change the interpretation of the current data set.

[Answer]

We appreciate for your valuable comment. We analyzed the DC subtypes in tumor-draining LNs and tumor tissues from B16F10-Ova tumor-bearing mice treated with Y27632 or vehicle. We found that CD103+ DCs were increased in tumor-draining LNs and tumors from Y27632-treated mice. We also found that cross-presentation of tumor antigen was increased in CD103+ DCs and CD8+ DCs from tumor-draining LNs as well as CD103+ DCs from tumors from Y27632-treated mice. Because migratory CD103+ DCs are the main myeloid population for tumor antigen uptake and trafficking, we isolated CD11c+ DCs from tumors and analyzed the ability of CD103+ DCs to phagocytose B16F10-Ova cells. The results showed that CD103+ DCs, but not CD11b+ DCs efficiently engulfed tumor cells. Furthermore, we found that CD40 expression was increased in CD103+ DCs from Y27632-treated mice compared with those from vehicle-treated mice (Fig. 4i). These findings suggest that increased phagocytosis by CD103+DCs could increase DC maturation and cross-presentation of tumor antigen, leading to increase in T-cell priming and antitumour immunity. These results were incorporated into Fig. 4g-j and Supplementary Figs. 12-15.

The description for the result was added in Results section (p.9, lines 16 ~ p.10, line 9) and Discussion section (p. 15, lines 3-13).

The experimental procedures were added in Methods section (p. 20, line 23 ~ p.21, line 2; p. 23, lines 13-19; p.24, lines 21-25; p.26, lines 7-11).

Finally, although combination of Y27632 and doxorubicin had obviously strong therapeutic effect in cancer model, their effects appeared to be additive rather than synergistic.

Investigation of the effect of combination therapy of Y27632 with other cancer drugs of different mechanisms, e.g. checkpoint inhibitor anti-PD1 Ab, maybe helpful to further clarify this issue.

[Answer]

To assess the importance of our combined therapy on antitumour immunity, we analyzed the therapeutic effect of combination of ROCK inhibitor and a non-ICD inducer. Cisplatin was used as a non-ICD inducer. We found that 3 mg/kg of cisplatin exerted a similar anti-tumour effect with 5 mg/kg of doxorubicin in B16F10-Ova tumour-bearing nude mice. Treatment with cisplatin was lower therapeutic effect than doxorubicin in immunocompetent mice, suggesting the role of ICD inducer in anti-tumor immunity. Although monotherapy of cisplatin and Y27632 decreased tumor burden to 54% and 32%, respectively, no additive effect on the reduction of tumor size was found in combination of cisplatin and Y27632. Moreover, any additive effect on DC maturation and cross-presentation was not found in combination of Y27632 and cisplatin. These findings suggest that combination with doxorubicin contributes to potentiation of Y27632-mediated anti-tumor immunity.

These results were incorporated into Supplementary Fig. 21.

The description for the result was added in Results section (p.12, lines 6-13) and Discussion section (p. 16, lines 18-21).

The experimental procedures were added in Methods section (p. 22, lines 4-5).

Major comments

1) The authors concluded from Fig 1h and Fig S3 that ROCK blockade augments phagocytosis by modulating phagocytes rather than by acting on tumor cells, because only the pre-treatment of Y27632 with phagocytes but not tumor promotes the phagocytosis ability. Given that Y27632 is a reversible inhibitor (Narumiya et al., *Methods Enzymol* 325, 273-284, 2000) and rapidly recover upon washout in vitro within few minutes (for example, see Nakazawa et al., *PNAS* 113, E6813-E6822, 2016), it is surprising that pre-treatment of phagocytes with Y27632 for 2 hours could strongly suppress the phagocytosis activity. Had the authors ever checked that the effect of pretreatment of Y27632 on phagocytes for 2 hours long-lasting or reversible? If the effect is long-lasting, there is a possibility that Y27632 change the phagocytes characteristic (e.g. cell differentiation) critical for their phagocytosis

ability rather than directly regulate the phagocytosis process. It is necessary to clarify this important issue.

[Answer]

As the reviewer commented, it has been known that Y27632 is a reversible inhibitor for Rho-kinase. In a suggested reference, the Y27632-mediated effect was reduced to 50% at 30 minutes after drug withdrawal. To assess how long the phagocytosis-promoting effect of Y27632 lasts after washout, we started phagocytosis assays at 0, 0.5, 1, 2, 4 hours after Y27632 washout and analyzed phagocytosis (%) after 2-hour incubation. After Y27632 washout, phagocytosis-promoting effect was declined with time. When tumour cells were added at 2 hours after Y27632 washout, phagocytosis (%) was reduced to the value observed in untreated cells. However, the percentages of phagocytosis at 0 hour after washout were similar to that of cells without washout. At 30 minutes, phagocytosis (%) was reduced by 15-30% in BMDMs and 3-5% in BMDCs. Although the effect of Y27632 was reversible, it was slowly declined. This might be because phagocytosis (%) was estimated after 2-hour incubation in our study.

The result is incorporated in Supplementary figure 3.

The description for the result is added in Results section (p. 5, lines 16-18).

The experimental procedure is added in Methods section (p. 20, lines 12-14).

2) The authors showed in Fig 4 that although there was no significant change in the maturation of DCs, Y27632 treated-CD11c+ DCs isolated from tumor-draining LNs and tumor significantly promotes antigen-specific T cell activation as evident by IFN gamma production. It is clear that T-cell priming by DCs is important for eliciting an antitumor immune response after Y27632 therapy but what is the mechanistic explanation for this Y27632-treated DCs T-cell priming ability? Is there a direct causal-relationship between DCs T cell priming ability and the increased phagocytosis activity of DCs they reported in Fig 1 and Fig 2? In addition, analyses of the DCs subtypes in tumor-draining LNs and tumor are needed because the changes affect antitumor immune response. Did the authors observe any increase or decrease of particular DCs subtypes in their model upon Y27632 treatment?

[Answer]

We analyzed the subtypes of DCs in tumor-draining LN and tumor tissues from B16F10-Ova tumor-bearing mice treated with Y27632 or vehicle. We found that CD103+ cDCs were increased in tumor-draining LN and tumors from Y27632-treated mice. We also showed that cross-presentation of tumor antigen was increased in CD103+ DCs and CD8+ DCs, but not CD11b+ DCs from tumour-draining LNs from Y27632-treated mice. Because migratory CD103+ DCs are the main myeloid population for tumor antigen uptake and trafficking, we isolated CD11c+ DCs from tumors and analyzed the ability of CD103+ DCs to phagocytose B16F10-Ova cells. The results showed that migratory CD103+ DCs, but not CD11b+ DCs efficiently engulfed tumor cells. Furthermore, we found that CD40 expression was increased in CD103+ DCs from Y27632-treated mice compared with those from vehicle-treated mice (Fig. 4i). These findings suggest that increased phagocytosis by CD103+DCs could increase maturation and cross-presentation of tumor antigen, leading to increase in T-cell priming and antitumour immunity.

These results are incorporated into Fig. 4g-j and Supplementary Figs. 12-15.

The description for the result was added in Results section (p.9, lines 16 ~ p.10, line 9) and Discussion section (p. 15, lines 3-13).

The experimental procedures were added in Methods section (p. 20, line 22 ~ p.21, line 2; p. 23, lines 13-19; p.24, lines 21-25; p.26, lines 7-11).

3) In Fig 6, the authors obviously showed that combination of Y27632 and an ICD inducer efficiently suppresses tumor growth. However, although only the combination of Y27632 and Dox could significantly induce DCs maturation as demonstrated in Fig 6c & 6d and be considered as synergistic effect, its importance on antitumor immunity is unclear because the effect on 3 important parameters of antitumor immunity including, the reduction of tumor size, percentage of H-2Kb-OVA+ DCs and IFN gamma production by antigen-specific OT-I T

cells, look additive rather than synergistic. It may be helpful to see the therapeutic effect of the combination of Y27632 with other anti-cancer drugs, such as anti-PD1 antibody and compare the results with the current findings.

[Answer]

We are grateful to the reviewer for this important comment. As the reviewer mentioned, the effect of combined therapy on the reduction of tumor size is additive. To assess the importance of our combined therapy on antitumor immunity, we analyzed the therapeutic effect of combination of ROCK inhibitor and a non-ICD inducer. Cisplatin was used as a non-ICD inducer. We found that 3 mg/kg of cisplatin shows a similar anti-tumor effect with 5 mg/kg of doxorubicin in B16F10-Ova tumor-bearing nude mice. Treatment with cisplatin was lower therapeutic effect than doxorubicin in immunocompetent mice, suggesting the role of ICD inducer in anti-tumor immunity. Although monotherapy of cisplatin and Y27632 decreased tumor burden to 54% and 32%, respectively, no additive effect on the reduction of tumor size was found in combination of cisplatin and Y27632.

Furthermore, Y27632 and Dox monotherapy increased H-2Kb-OVA+DCs by 130.2% and 39.3% compared with vehicle-treated control, respectively, whereas combined therapy increased by 321.5% (Fig. 6d). In Fig. 6e, Y27632 and Dox monotherapy increased IFN-gamma production by 160.5% and 45.5% compared with vehicle, respectively, whereas combined therapy increased by 333.2%. In Fig. 6f, Y27632 and Dox increased the percentage of IFN gamma-positive cells by 67.3% and 20.8%, respectively, whereas combined therapy increased by 164.1%. These results showed that the effects of combined therapy on these parameters are more than the additive (albeit only modestly). However, any additive effect on DC maturation and cross-presentation was not found in combination of cisplatin and Y27632. These findings suggest that ICD inducer contributes to enhancement of Y27632-mediated anti-tumor immunity.

These results are incorporated into Supplementary Fig. 21.

The description for the result was added in Results section (p.12, lines 6-12) and Discussion section (p. 16, lines 18-21).

The experimental procedures were added in Methods section (p. 22, lines 1-3).

Because the effect of combined therapy was modestly synergistic, “synergistically” or “synergism” was changed into “markedly” or other expressions (p.13, line 9; p. 16, line 18 and line 22; p.17, line 6)

Minor comments

1) Fig 2d. I'm not sure about the statistics significance, but clodronate treatment reduced the tumor size. This finding may suggest the existence of immune suppression macrophages in this model. Could the authors add a discussion about this issue in their manuscript?

[Answer]

As shown in Figure 2d, clodronate treatment partially reduced tumor size, as compared to PBS liposome-treated mice, however, this reduction is not statistically significant.

PBS liposome, PBS Vs Clodronate liposome, PBS: p-value=0.098

PBS liposome, PBS Vs clodronate liposome, Y27632: P-value=0.138

As the reviewer mentioned, tumour-associated macrophages can play a protumoral role through different mechanisms. Thus, it is possible that immune suppressive macrophages exist in this model.

The issue was described in Discussion section (p. 14, line 21 ~ p.15, line 1).

2) Fig4a. The author presented their data as percentage of cells in tumor-draining LNs, but I'm wondering what about the absolute cell numbers. Although the percentage is not significantly different, is it possible that the absolute cell number is larger in the Y27632-treated group and therefore results in stronger antitumor immunity.

[Answer]

We analyzed the absolute cell number of CD40+ and CD86+ DC in tumor-draining LNs from B16F10-ova tumor-bearing mice. Absolute cell number of CD40+ and CD86+ DCs was slightly increased in Y27632-treated mice, but this increase was not statistically significant.

This result is incorporated into Supplementary Fig. 11c.

On the other hand, analysis of DC subtypes revealed that cross presentation and phagocytosis was efficiently increased in CD103+ DCs from Y27632-treated mice. This finding led us to examine DC maturation in CD103+ DCs because the positive effect in small CD103+ DC population might be masked in analysis from total CD11c+ DCs. We found that CD40 expression was markedly increased in CD103+ DCs.

This result was incorporated into Fig. 4j.

The description for the result was added in Results section (p. 10, lines 4-7).

The experimental procedure was added in Methods section (p. 24, lines 21-25).

3) Fig 4d-e. The authors showed that Y27632 treatment of F4/80+ macrophages isolated and purified from TDLNs or tumors has no positive effect on T cell activation. However, as results in Fig 2d suggested the existence of inhibitory macrophages, it is possible that they are included in the purified F4/80+ macrophages used in this experiment and therefore mask the positive effect of Y-27632 in macrophages. Could the authors give some comments about this issue?

[Answer]

As the reviewer mentioned, the positive effect of Y27632 in macrophages may be masked by immune suppressed macrophages. The issue was described in Discussion section (p. 14, line 21 ~ p.15, line 1).

4) The results of Fig 5a together with Fig 1c suggested that Y27632 promotes BMDCs phagocytosis ability regardless of the nature of their target cells, live of Dox-treated ICD. Does the phagocytosis by Y27632-treated BMDCs has any selectivity against specific type of cells (or death cells)? Could Y27632-treatment also promote phagocytosis of target cells undergoing necrosis or non-ICD apoptosis?

[Answer]

We analyzed phagocytosis of cells dying a non-immunogenic and necrotic cell death in Y27632-treated BMDCs. We found that ROCK blockade increased their engulfment by BMDCs. This result indicates that Y27632 promotes phagocytic ability of BMDCs regardless of the nature of their target cells.

The result was incorporated in Supplementary figure 16.

The description for the result was added in Result section (p. 10, lines 18-20).

The experimental procedure was added in Method section (p. 20, lines 19-22).

5) Fig 5a. X axis should be corrected as BMDCs (CMFDA-DR).

[Answer]

Legend for X axis of Figure 5a was corrected as 'BMDCs (CellTracker Deep Red)'. Legend for Y axis of Figure 5a was also corrected as 'Tumour cells (CMFDA)'.

6) In the introduction of this paper, the authors cited Liu X et al. and Sockolosky JT et al. works and mentioned that "therapeutic agents that antagonize CD47, which binds to the receptor, SIRPa (signal-regulatory protein alpha), on macrophages and simulates anti-phagocytic signaling, have been found to drive T cell mediated elimination of immunogenic tumours". According to these 2 works, it was shown that blockage of CD47 results in increased phagocytosis in both macrophages and DCs. In addition, it was also shown that CD47-specific blocking involved in the T cell mediated elimination of immunogenic tumors. However, according to the latest related-work (Xu et al., Immunity 47, 363-373, 2017), the underlying mechanism of anti-tumor effect by CD47 blocking is largely depended on the sensing of mtDNA in the DCs subtype producing type I interferon but not the increased phagocytosis ability of macrophages and DCs. The authors should also cite this work.

[Answer]

In page 3, line 8, "macrophages" was changed into "macrophages and DCs".

The description for the suggested reference (Xu et al., Immunity 47, 363-373, 2017) was added in Introduction section (p.3, lines 11-14), and the reference was cited (ref. 4).

7) What are the identity of the gray peaks in Fig 2e, Fig 5e, Supplementary Fig 1, Supplementary Fig 6, Supplementary Fig 7 and Supplementary Fig 8? I couldn't find explanations in the figure legends. Are they the staining with isotype control?

[Answer]

In Fig. 2e, the gray peaks indicate the mCherry-positive signal in DCs or macrophages from CT26.CL25 tumor-bearing mice (mCherry-negative tumor).

In Fig. 5e, Supplementary Fig. 1, Supplementary Fig. 6, Supplementary Fig. 7, and Supplementary Fig. 8, the gray peaks indicate the staining with isotype-matched control IgG. Description for the gray peak was added in each figure legend.

Information of isotype control antibodies was added in Methods section.

8) Several published works suggested the inhibitory role of Y-27632 on DC migration and therefore immunosuppression (e.g. Lammermann et al., Nature 453, 51-55, 2008; Nitschke et al., Blood 120, 2249-2258, 2012). How could the authors' findings in the present work reconcile with these previous findings? Could the authors provide some discussions on this issue?

[Answer]

Dendritic cell migration to draining LNs is crucial for the initiation of adaptive immunity.

Suggested references showed that ROCK blockade reduced DC migration velocity.

However, we found that CD103⁺ DC increased in tumour-draining LNs from Y27632-treated mice. DCs rely on migration via stromal networks to carry antigens from parenchymal tissues to LNs (Turley SJ et al., Nat. Rev. Immunol. 10:813-825, 2010). A previous study showed that the reduction of actomyosin contractility via downregulation of RhoA activity and its downstream MLC phosphorylation contributed to CLEC-2-mediated DC migration along stromal cells scaffolds (Acton et al., Immunity, 37:276-89, 2012). Thus, migratory CD103⁺ DCs might preferentially use a specific mechanism that causes efficient migration into LNs such as CLEC-2 signaling.

The description for this issue was added in Discussion section (p.15, line 19 ~ p.16, line 2).

Reviewer #2 (Remarks to the Author):

The manuscript by Nam et al., which is well-written, presents evidence for the synergistic antitumour activity of a chemotherapeutic (e.g., doxorubicin) and rho kinase (ROCK) inhibitor in implanted as well as spontaneous tumors. The initial focus is on ROCK inhibition as a stand-alone analysis. In vitro, Y27632 treatment of BMDM or BMDC cultured with either the CT26.CL25 (colon cancer) or B16F10-Ova (melanoma) tumour lines resulted in enhanced phagocytosis of the tumour cells. However, looking at the numbers that the authors provided regarding percent of phagocytosis, it was unclear exactly how it was calculated. It is mentioned in the Methods, but my calculations come out differently.

[Answer]

We analyzed cancer cell phagocytosis using two methods: flow cytometry analysis and microscopic analysis using pH sensitive dye.

In flow cytometry analysis, phagocytosis (%) was expressed as percentage of phagocytes engulfing cancer cell (right-upper quadrant) in total phagocytes (right quadrants). This formula was added in Methods section (p.20, lines 10-12).

In Fig. 1b, average phagocytosis (%) for CT26.CL25 and B16F10 cells were 3.76 or 3.1 in vehicle-treated-BMDMs and 9.37 or 8.43 in Y27632-treated BMDMs, respectively. In Fig. 1a (a representative result), phagocytosis (%) were 2.65 $(1.79/(1.79+65.7) \times 100)$ and 2.63 $(1.82/(1.82+67.5) \times 100)$ in vehicle and 10.33 $(7.45/(7.45+64.7) \times 100)$ and 7.24 $(5.05/(5.05+64.7) \times 100)$ in Y27632, respectively.

In Fig. 1d, average phagocytosis (%) for CT26.CL25 or B16F10 cells were 2.92 and 4.52 in vehicle-treated BMDCs and 9.99 and 10.09 in Y27632-treated BMDCs, respectively. In Fig. 1c (a representative result), phagocytosis (%) were 3.03 $(1.23/(1.23+39.3) \times 100)$ and 3.59

$(2.02/(2.02+54.3) \times 100)$ in vehicle and $11.14 (5.18/(5.18+41.3) \times 100)$ and $10.53 (5.98/(5.98+50.8) \times 100)$ in Y27632, respectively.

In Fig. 5b, average phagocytosis (%) for ICD-induced CT26.CL25 or B16F10 cells were 16.6 and 21.0 in vehicle-treated BMDCs and 29.1 and 36.8 in Y27632-treated BMDCs, respectively. In Fig. 5a (a representative result), phagocytosis (%) were 15.9 $(8.27/(8.27+43.9) \times 100)$ and 21.9 $(13.8/(13.8+49.1) \times 100)$ in vehicle and 30.2 $(15.2/(15.2+35.1) \times 100)$ and 38.3 $(21.8/(21.8+35.1) \times 100)$ in Y27632, respectively.

In microscopic analysis using pH-sensitive dye, phagocytosis (%) was calculated as percentage of phagocytes engulfing cancer cell (yellow) in total phagocytes (green) in random microscopic fields.

In Fig. 1g, average phagocytosis (%) for CT26.CL25 cells were 1.35 and 8.84 in vehicle-treated and Y27632-treated BMDMs. In Fig. 1e (a representative result), phagocytosis (%) were 1.72 $(1/58 \times 100)$ and 9.75 $(4/41 \times 100)$ in both cells, respectively.

In Fig. 1g, average phagocytosis (%) for CT26.CL25 cells were 1.38 and 6.7 in vehicle-treated and Y27632-treated BMDCs, respectively. In Fig. 1f (a representative result), phagocytosis (%) were 2.32 $(3/129 \times 100)$ and 8.42 $(8/95 \times 100)$ in both cells, respectively. In Fig. 5d, average phagocytosis (%) for ICD-induced CT26.CL25 cells is 9.98 and 20.17 in vehicle-treated and Y27632-treated BMDCs, respectively. In Fig. 5c (a representative result), phagocytosis (%) were 13.6 $(33/241 \times 100)$ and 23.3 $(73/313 \times 100)$ in both cells, respectively.

Most of the in vitro experiments that you would want to see were there, as were the appropriate in vivo challenge and spontaneous tumour model systems; the appropriate statistical analyses were performed. Unfortunately, particularly for the in vitro experiments, certain controls were not always included.

[Answer]

We are grateful to the reviewer for this important comment. In supplementary Fig. 1, expression of F4/80 and CD11c was analyzed in bone marrow cells before differentiation induction, and this data was incorporated into Supplementary Fig. 1.

In Supplementary Fig. 2, the effect of Y27632 on tumor cell apoptosis was analyzed including positive controls (mouse TRAIL and doxorubicin) (specific question 2).

In Fig. 5e, Supplementary Fig. 1, 6, 7, and 8, (Supplementary Fig. 1, 8, 10, 11 in the revised manuscript), the gray peaks indicate the staining with isotype-matched control IgG.

Information for the gray peak was added in each figure legend.

In Fig. 5g, the results from isotype control for anti-H2Kb-OVA antibody were provided in Supplementary Fig. 18.

Although the data are certainly consistent with the authors premise, there are some holes that do not fully pin down the mechanism of antitumour activity. This has reduced this reviewer's overall enthusiasm for this manuscript.

[Answer]

We analyzed the subtypes of DCs in tumor-draining LN and tumor tissues from B16F10-Ova tumor-bearing mice treated with Y27632 or vehicle. We found that CD103+ cDCs were increased in tumor-draining LN and tumors from Y27632-treated mice. We also showed that cross-presentation of tumor antigen was increased in CD103+ DCs and CD8+ DCs, but not CD11b+ DCs from tumour-draining LNs from Y27632-treated mice. Because migratory CD103+ DCs are the main myeloid population for tumor antigen uptake and trafficking, we isolated CD11c+ DCs from tumors and analyzed the ability of CD103+ DCs to phagocytose B16F10-Ova cells. The results showed that migratory CD103+ DCs, but not CD11b+ DCs efficiently engulfed tumor cells. Furthermore, we found that CD40 expression was increased in CD103+ DCs from Y27632-treated mice compared with those from vehicle-treated mice (Fig. 4i). These findings suggest that increased phagocytosis by CD103+DCs could increase maturation and cross-presentation of tumor antigen, leading to increase in T-cell priming and antitumour immunity.

These results are incorporated into Fig. 4g-j and Supplementary Figs. 12-15.

The description for the result was added in Results section (p.9, lines 16 ~ p.10, line 9) and Discussion section (p. 15, lines 3-13).

The experimental procedures were added in Methods section (p. 20, line 23 ~ p.21, line 2; p. 23, lines 13-19; p.24, lines 21-25; p.26, lines 7-11).

Specific questions, comments and criticisms:

1. Supplemental Fig. 2 is meant to show that the ROCK inhibitor Y27632 does not induce apoptosis in the tumour cells. An important positive control would be treating the tumour cells with a drug that definitely causes activation-induced cell death.

[Answer]

As the reviewer mentioned, recombinant mouse TRAIL was used as a positive control that definitely causes activation-induced cell death (AICD). However, B16F10-Ova melanoma and CT26.CL25 colon cancer cells was resistant for TRAIL-induced apoptosis. Especially CT26.CL25 cells were completely resistant for 10 ng/ml recombinant TRAIL. These findings are in agreement with the previous results (Kayagaki N et al., J Immunol 163:1906-13, 1999; Liu F et al., PLoS One 6:e16241, 2011). Thus, we also used doxorubicin as a positive control.

The result was incorporated to Supplementary Fig. 2.

2. In Fig. 2, macrophages and DC from Y27632-treated tumour-bearing mice were better at phagocytosing tumour cells than were those from vehicle-treated tumour-bearing mice. Does Y27632 itself enhance phagocytic activity in these cell types from mice that are not bearing tumours?

[Answer]

To investigate whether Y27632 enhance phagocytic activity in macrophages and DCs from mice that are not bearing tumors, thymocytes were isolated from 8-week-old mice and induced apoptosis by incubating with dexamethasone. Then, CFSE-stained, apoptotic thymocytes were injected (i.v.) into mice treated with Y27632 or vehicle as indicated in Supplementary Fig. 9a. After 1 hour, F4/80+ macrophages and CD11c+ DCs were isolated from spleen, and phagocytosis (%) was analyzed. The results showed that Y27632 increased phagocytosis of apoptotic thymocytes in splenic macrophages and DCs in mice that are not bearing tumors.

The results were incorporated into Supplementary Fig. 9.

Description for this result was added in Results section (p. 7, lines 11-12).

The experimental procedure was added in Methods section (p. 21, lines 11-15).

3. Which cells are making IFN-gamma in the draining LNs? How do you know?

[Answer]

The primary sources of IFN-gamma are natural killer (NK) cells and natural killer T (NKT) cells, which are effectors of the innate immune response, and CD8 and CD4 Th1 effector T cells of the adaptive immune system. Among them, cytotoxic CD8+ T cells are activated by signals from the TCR and costimulatory molecules in response to cognate MHC class I:peptide complexes (Schoenborn JR et al., Adv Immunol. 2007;96:41-101). To investigate whether CD8+ T cells are responsible for Y27632-mediated IFN-gamma production in the tumour-draining LN cells in response to b-gal peptide, we isolated CD8+ and CD8- cells from tumor-draining LN cells from B16F10-Ova tumor-bearing mice. Then, each cell population was incubated with b-gal peptide and analyzed IFN-gamma production. We found that CD8+ T cells were the main cells for making IFN-gamma in tumor-draining LNs.

This result was incorporated into Fig. 2h.

Description for these results was added in Results section (p. 7, lines 18-24).

The experimental procedure was added in Methods section (p. 24, lines 3-5).

4. In the experiment presented in Fig. 5g, the authors argue that "Cross-presentation of an OVA-derived peptide to MHC-I was markedly increased by co-culture of BMDCs with Dox-treated B16-F10-Ova cells in the presence of Y27632..." Actually, without Dox, there is a

greater increase in cross-presentation (2.2-fold increase vs. 1.66-fold with Dox). This would suggest Dox does not enhance the cross-presentation of the Kb/OVA complex.

[Answer]

As the reviewer mentioned, there is a greater increase in cross-presentation when BMDCs were co-cultured with tumor cells not treated Dox in vitro. Although cross-presentation is increased in BMDCs co-cultured with Dox-treated tumor cells versus BMDCs co-cultured with live tumor cells, this increase may be due to increased phagocytosis. Moreover, cross-presentation was not statistically increased in DCs from tumor-bearing mice treated with Dox monotherapy (Fig. 6d). Thus, it is possible that Dox does not directly enhance the cross-presentation of the Kb/OVA complex.

“we found that ROCK blockade markedly enhanced DC-mediated phagocytosis of cell dying an immunogenic death in response to Dox and cross-presentation of cancer cell-specific antigen compared to each monotherapy” was changed into “we found that ROCK blockade markedly enhanced DC-mediated phagocytosis of cell dying an immunogenic death in response to Dox” (p.16, lines 11-12).

However, combination of Y27632 and Dox cooperatively increased cross-presentation of tumor antigen compared with Y27632 or Dox monotherapy (Fig. 6d), suggesting that Dox treatment contributed to enhancement of cross-presentation in Y27632-treated DCs.

Furthermore, we analyzed therapeutic effect of combination of Y27632 and a non-ICD inducer (cisplatin). Any additive effect on cross presentation was not found in combined therapy using Y27632 and cisplatin (Supplementary Fig. 21). This result suggests that some factors from cells dying an immunogenic cell death contribute to increased cross-presentation by combined therapy.

The descriptions related to this issue were revised in Discussion section (p. 16, lines 12-14 and 17-21).

5. In Fig. 5e, co-culture of BMDCs with Dox-treated tumour cells had enhanced (albeit really only modestly) CD40 and CD86 surface expression. Fig. 4 looks at the percentage of BMDCs that are CD40+ and CD86+ and Y27632 treatment made no difference, but no analysis of surface expression by flow cytometry was apparently done. What effect does Y27632 treatment have on co-stimulatory molecule expression on BMDCs co-cultured with tumour cells not treated with Dox?

[Answer]

In this study, we had analyzed the percentages of CD40+ and CD86+ cells in BMDCs co-cultured with untreated (Supplementary Fig. 8b,c) or Dox-treated tumor cells (Fig. 5e,f). (Supplementary Fig. 8b,c in original manuscript was changed into Supplementary Fig. 11d,e). For surface expression of co-stimulatory molecules, we provided the values of mean fluorescence intensity (MFI) in BMDCs.

These results are incorporated into Supplementary Fig. 11f and 17.

We had also analyzed the percentages of CD40 and CD86 in DCs from tumor-bearing mice treated Y27632 alone (Fig. 4a) or in combination with Dox (Fig. 6c). The relative MFI value for these experimental sets was also incorporated into Supplementary Fig. 11b and 19.

Taken together, we did not find a significant difference in CD40 and CD86 expression in DCs from tumor-draining LNs, except to combination of Y27632 and Dox.

On the other hand, we found that CD103+ DC population efficiently increased phagocytosis (Fig. 4j) and cross-presentation of tumor antigen by Y27632 treatment (Fig. 4h). Because we thought that positive effect in a DC subset may be not detected in the experiment from total tumor-draining LN cells, we analyzed CD40 expression in CD11c+CD103+ DC population. We found that CD40 expression was increased in CD103+ DCs from Y27632-treated mice compared with those from vehicle-treated mice (Fig. 4i).

Description for this result was added in Results section (p. 10, lines 4-7).

6. Fig. 7c shows that tumour-bearing mice treated with both Dox and Y27632 had 100% survival. This is remarkable. How many times was this experiment performed? Did you

always get 100% survival with this dual treatment regimen?

[Answer]

This experiment was performed two times (n=7 per group), and the mice with combined therapy had 100% survival at 50 days after treatment. Survival events were scored when tumor burden reached 2,500mm³ or per absolute survival events.

Minor question:

1. Kb binds peptides that are 8 or 9 amino acids in length. The Neu-derived peptide is a 10-mer. Is this a common sequence for this molecule in FVB (H-2q) mice? It was not cited.

[Answer]

Rat-Neu (420-429) peptide was originally identified as an epitope expressed by the rat neu and recognized by FVB/N (H-2q)-derived CTL (Ercolini AM et al., J Immunol 2003). This reference was additionally cited (ref. 49) in Methods section (p.19, line 5). Many studies have used this peptide for characterization of the CD8+ T cell responses in the Neu breast cancer model (Ercolini AM et al., J Exp Med, 2005; Singh R et al., Cancer Res 2006; Gonzalez-Martin A et al., Cancer Res 2011).

Others]

1. Supplementary Fig. 3, 4, 5, 6, 7, 8, and 9 in original manuscript were changed into Supplementary Fig. 4, 6, 7, 8, 10, 11, and 18 in the revised manuscript, respectively.
2. Some errors were corrected and marked in the revised manuscript (marked manuscript). (p.19, line 11;
3. To prevent confusion between allophycocyanin (APC) and antigen-presenting cells (APC), APC was only used as an abbreviation for antigen-presenting cells. In Method section and figure legends, "APC" was changed into "allophycocyanin".
4. In all figures, "tumor" was changed into "tumour".
5. "Yoonjeong Choi" was added as a co-author.
6. Requirements from Nature Communications checklists were added in Methods section (p.19, line 25 ~ p.20, line 1; p.22, lines 10-11; p.26, lines 22-23; p.27, lines 10-12).
7. Reagents used during the revision was added in Methods section (Reagents subsection).

Reviewers' comments:

Reviewer #1 (Remarks to the Author):

The revised manuscript by Nam, et al. is greatly improved. The authors addressed the reviewer's comments with appropriate text editing and new experiments. I found that analysis on DCs subsets is particularly helpful. However, I still have some concerns on the following issues;

1) I'm wondering about the author's explanation for their Figure 6c data. The authors showed that only Y27632+Dox treatment increased the maturation of total DCs in tumor-draining LN. Why? Although the authors mentioned that "ICD of tumor cells by Dox contributes to potentiation of Y27632-mediated antitumour immunity" as a possible mechanism underlying the "markedly" augmented DC maturation by the combination Y27632 and ICD inducer (Dox) in the "Discussion" on page 16 line 20-21, they should be aware that CD103+ DCs is a small population of total DCs in tumor and tumor-draining LN (as they mentioned in their responses to my second minor comment). Therefore, it is unlikely that increased maturation of CD103+ DCs population (which is observed upon Y27632 treatment) will be reflected in the total DCs. It would be helpful if the authors could further clarify their discussion on this important issue.

2) Regarding the authors response to my minor comment 6 about the relationship between anti-tumor effect of anti-CD47 treatment and phagocytosis, I think the authors revised introduction (page 3, line 6-11) is still ambiguous. Is it possible to describe more clearly? Moreover, I think the authors should also be aware that the anti-phagocytic role of CD47 is recently questioned (Nagata S, *Annu. Rev. Immunol.* 36:18.1-18.29, 2018) and the anti-tumor effect of anti-CD47 antibodies has been challenged (Horrigan SK, *eLIFE* 6, e18173, 2017). Therefore, anti-CD47 treatment might not be a good example of effort to modulate tumor cell phagocytosis for cancer immunotherapy.

Finally, I think a simplified cartoon or scheme may be helpful for general readers to quickly get through this comprehensive paper.

Reviewer #2 (Remarks to the Author):

The authors have made excellent changes to the original manuscript, being very responsive (in detail) to the reviewers' criticisms. They have also increased the number of Supplemental Figures to reinforce their response to the reviewers.

Response to Reviewers' comments:

Reviewer #1 (Remarks to the Author):

The revised manuscript by Nam, et al. is greatly improved. The authors addressed the reviewer's comments with appropriate text editing and new experiments. I found that analysis on DCs subsets is particularly helpful. However, I still have some concerns on the following issues;

1) I'm wondering about the author's explanation for their Figure 6c data. The authors showed that only Y27632+Dox treatment increased the maturation of total DCs in tumor-draining LN. Why? Although the authors mentioned that "ICD of tumor cells by Dox contributes to potentiation of Y27632-mediated antitumour immunity" as a possible mechanism underlying the "markedly" augmented DC maturation by the combination Y27632 and ICD inducer (Dox) in the "Discussion" on page 16 line 20-21, they should be aware that CD103+ DCs is a small population of total DCs in tumor and tumor-draining LN (as they mentioned in their responses to my second minor comment). Therefore, it is unlikely that increased maturation of CD103+ DCs population (which is observed upon Y27632 treatment) will be reflected in the total DCs. It would be helpful if the authors could further clarify their discussion on this important issue.

Answer]

In this study, increased expression of costimulatory markers was detectable in total CD11c⁺ DCs from tumour-draining LNs by only combination of Y27632 and Dox. As the reviewer commented, increased maturation of CD103+ DCs population by Y27632 was insufficient to explain this finding. On the other hand, our combined therapy might mediate DC maturation in several ways. First, considering that during uptake and processing of foreign antigens, immature DCs begin to mature and migrate to the spleen or adjacent lymph nodes, remarkable increase of tumour cell phagocytosis by DCs may accelerate DC maturation. Second, damage-associated molecular patterns (e.g. HMGB1) by immunogenic cell death could contribute DC maturation through activation of pattern recognition receptor signaling. Third, making immunosuppressive tumour microenvironments more immunogenic could also contribute to DC maturation. Thus, combination of these effects might cause detectable increase in DN maturation in total DCs from tumour-draining LNs. However, the precise mechanism by which combined therapy enhances DC maturation remains to be clarified.

The description for this issue was added in Discussion section (p.16, line 21 ~ p.17, line 3).

"Markedly" was deleted in Discussion section (p.16, line 18).

"HMPB1" was corrected as "HMGB1" (p.16, line 8).

2) Regarding the authors response to my minor comment 6 about the relationship between anti-tumor effect of anti-CD47 treatment and phagocytosis, I think the authors revised introduction (page 3, line 6-11) is still ambiguous. Is it possible to describe more clearly? Moreover, I think the authors should also be aware that the anti-phagocytic role of CD47 is recently questioned (Nagata S, Annu. Rev. Immunol. 36:18.1-18.29, 2018) and the anti-tumor effect of anti-CD47 antibodies has been challenged (Horrigan SK, eLIFE 6, e18173, 2017). Therefore, anti-CD47 treatment might not be a good example of effort to modulate tumor cell phagocytosis for cancer immunotherapy.

Answer]

As the reviewer commented, recent studies showed that anti-CD47 treatment might not be a good example of effort to modulate tumor cell phagocytosis for cancer immunotherapy. Thus, descriptions for CD47 were deleted from Introduction section. Description for anti-CD20 antibody and related references were added into Introduction section (p.3, lines 6-8) because Ab-dependent cellular phagocytosis by Kupffer cells was identified as a primary mechanism of anti-CD20 therapy (ref 2).

Finally, I think a simplified cartoon or scheme may be helpful for general readers to quickly get through this comprehensive paper.

Answer]

A simplified scheme for our study was added and incorporated into Figure 8.

“Fig. 8” was added in p.14, line 16.

REVIEWERS' COMMENTS:

Reviewer #1 (Remarks to the Author):

I have no further comments. I think this interesting study is now ready for publication.